# Bystanders of intimate partner violence against women and their willingness to intervene: An analysis of secondary data in Spain (2005–2020)

**Andrés Sánchez-Prada**[1⊙], **Carmen Delgado-Alvarez**[1⊙], **Esperanza Bosch-Fiol**[2⊙], **Virginia Ferreiro-Basurto**[2⊙], **Victoria A. Ferrer-Perez**[2⊙]*

**1** Faculty of Psychology, Pontifical University of Salamanca, Salamanca, Spain, **2** Faculty of Psychology, University of Balearic Islands, Palma, Spain

⊙ These authors contributed equally to this work.
* victoria.ferrer@uib.es

## Abstract

Recent decades have seen a growing acknowledgement of violence against women (VAW) as a serious social and public health problem of epidemic proportions. The prevention of VAW and intimate partner violence against women (IPVAW) has become a priority within this context, and includes various prevention strategies such as social participation and helping behaviors. In different countries, conducting research on help-seeking behavior and bystander intervention in cases of VAW is a common practice, but addressing these issues is much less common in Spain. In this context, the objective of this study is to provide a preliminary estimation of the volume of bystanders in cases of IPVAW in Spain between 2005 and 2020 (since the entry into force of Organic Law 1/2004), their willingness to intervene and, in the case of intervention, the type of helping behavior (real or hypothetical) preferred, using the sources (secondary data) available (specifically, survey data, as the surveys of social perception of gender violence and the 2014 and 2019 macro-surveys, and also administrative data, as the database of reports filed). The data analyzed allow us to determine that, in fact, in the cases of IPVAW there are usually persons within the victim's inner circle who are firsthand witnesses or have been informed by the victim of the existence of this type of violence, but, although the bystanders generally claim they would engage in an active and supportive response, this is in fact not always the case. These results underscore the need to develop intervention programs aimed at IPVAW bystanders to improve their reaction and contribute to the development of helpful and efficient active responses.

## Introduction

Recent decades have seen a growing acknowledgement of violence against women (VAW) as a serious social and public health problem of epidemic proportions among different world organizations (United Nations, World Health Organization) and the governments of numerous

---

**Data Availability Statement:** The methodological strategy used was an analysis of secondary data obtained and processed by other sources. In this

case, quantitative data taken from sociological surveys conducted in Spain by different institutions and administrative data related to the records of complaints for intimate partner violence against women and they are fully available on the web freely for anyone who wishes to consult them. The URLs from the Centro de Investigaciones Sociologicas (https://www.cis.es/cis/opencms/EN/index.html) where the quantitative data from sociological surveys are available are: - Social perception of gender violence (study 2968): https://analisis.cis.es/cisdb.jsp?ESTUDIO=2968 - Social perception of gender violence among adolescents and youth (study 2992): https://analisis.cis.es/cisdb.jsp?ESTUDIO=2992 - Macro-survey on violence against women 2014 (study 3027): https://analisis.cis.es/cisdb.jsp?ESTUDIO=3027 - Macro-survey on violence against women 2019 (study 3235): https://analisis.cis.es/cisdb.jsp?ESTUDIO=3235 The URL from the Statistics Portal of the Spanish Government Delegation for Gender Violence where the administrative data related to the records of complaints for intimate partner violence against women are available are: http://estadisticasviolenciagenero.igualdad.mpr.gob.es/ All these data are fully anonymized.

**Funding:** This work has been financed by the State Research Agency (Agencia Estatal de Investigación, AEI) and the Ministry of Economy and Innovation (Ministerio de Economía y Innovación, MCIN) through the Research Project PID2019-104006RB-I00/, funded by MCIN / AEI / 10.13039/501100011033.

countries [1, 2]. Among the various types of VAW, intimate partner violence against women (IPVAW) is the most common [3–6].

The prevention of VAW and IPVAW has become a priority within this context [7], and includes various prevention strategies such as social participation and helping behaviors [8, 9]. As in the case of other emergency situations, bystanders of VAW and IPVAW have the option of being either "passive" or "inactive", meaning they may choose not to become involved, to ignore the situation and/or keep quiet and do nothing, supporting the perpetrator and/or blaming the victim, or "active", meaning to engage and intervene to help the victim providing different forms of informal support to the them (for example, offering assistance, helping them to make decisions, talking to the them, helping or accompanying her to access support services, or helping her to report the case to the police) and/or stop the violence (for example, taking personal action, confronting the aggressor, requesting legal intervention, or reporting the case to the police) [10–18], thus contributing, or not, to the prevention of such acts of abuse.

Given the importance of this bystander approach in preventing and confronting any kind of VAW [8–11, 19], the focus of this paper is bystanders' behavior and their willingness to intervene in cases of VAW in Spain, understood in the theoretical context of Latané and Darley's [20] bystander model of intervention in emergencies, and of Banyard' adaptation [21] to the specific characteristics of VAW.

It is important to note that, although some authors (e.g., Pease & Flood [22] or Walter-maurer [23]) consider the entire community to be witnesses or bystanders of VAW, others (e.g., EIGE [17]; Herrero et al. [24]) distinguish a witness or bystander (understood as any non-professional adult who observes, suspects, or is otherwise made aware of VAW, including acquaintances, family members, friends, colleagues or neighbors) from professionals who are aware of this violence (i.e., police, court members, health and social care workers, or workers from specialized victim services). Our research assumes this distinction and focuses on non-professional witnesses or bystanders [17]. In relation to this issue, it is important to highlight that, although VAW are generally considered, to a certain extent, 'hidden' crimes in the eyes of justice and society [25, 26], very often there are a significant number of people within the personal or inner circle of the female victims of these violence who, in one way or another, are witness to the perpetration of violence [11, 14, 19, 27, 28], and the women often turn to this inner circle for help [27–30]. In different countries, conducting research on help-seeking behavior and bystander intervention in cases of VAW is a common practice and has been carried out for various forms of VAW such as dating violence, sexual assault, or sexual violence [10, 14, 15, 31–33], although reliable data on their effectiveness is not available for all cases [34, 35]. Addressing these issues is much less common in Spain [11, 36], with most research being circumscribed to IPVAW [11, 12, 36, 37].

In this sense, a scoping review on bystander helping behaviors in VAW cases in Spain [36] shows that there is indeed a limited amount of research on this topic, but research carried out suggest that a substantial percentage of people have been witness in VAW cases. Thus, for instance, some sociological surveys show that between 18% and 23% of Spanish adults know a woman victim of IPVAW in their circle of friends or family [27, 28, 38–46]; about 36% of adolescents have knowledge of couples in which the boy abuses the girl [47–50]; about 27% of Spanish adults know a woman victim of sexual assault [51]; and among university students, 62% have either experienced directly or know someone who has experienced some type of gender-based violence at university [52]. Moreover, as previous research in many countries points out (i.e., Lazarus & Signal [53]), Spanish women have generally reported a greater knowledge of different VAW cases in IPVAW, sexual assault or other types of VAW than Spanish men [43, 49–51].

Related to the bystander behavior, reporting witnessed violence or encouraging victims to report the violence suffered are some of the most common behaviors identified by previous

Spanish research on the topic, among both the general adult population, adolescents and university students [40–42, 44, 49, 50, 54]. It is relevant to point out that, in general, women were found to encourage victims to report the situation of violence to someone who can punish the aggressor. Men, on the other hand, were more likely to confront the aggressor directly or to make passive responses [43, 44, 48, 55–57]. Additionally, some studies [11, 58–60] point out the existence of a relevant effect of age on the intention to help, although the meaning of this relationship has not been fully supported.

In this context, the objective of this study is to provide a preliminary estimation of the volume of bystanders in cases of IPVAW in Spain between 2005 and 2020 (since the entry into force of Organic Law 1/2004, a specific law related to IPVAW), their willingness to intervene and, in the case of intervention, the type of helping behavior (real or hypothetical) preferred. Although, as mentioned above, there is a limited amount of previous research on this topic in Spain [11, 36], we have identified some secondary sources of data regarding IPVAW that provide relevant information about the number of bystanders and their involvement (real or hypothetical) in these cases [36]. That is why this study uses these sources (survey data, as the surveys of social perception of gender violence [43, 49] and the 2014 and 2019 macro-surveys [40–42]; and administrative data, as the database of reports filed [61]), to broader our knowledge of this topic. Additionally, and given that the results in the literature on this matter suggest the potential effect on the response by variables such as sex [43, 44, 48–51, 55–57] and age [12, 13, 19, 58–60], these relationships will be analyzed in greater detail.

As mentioned before, one of the bystander behaviors to support IPVAW victims is to report the case to the police. This type of information is readily available in Spain. In fact, the Statistics Portal of the Government Delegation for Gender Violence [61] has gathered and provided information on the number of complaints of IPVAW annually (since 2009), per quarter, per territory, and also per source of complaint.

Psychosocial studies on the matter have usually focused on an analysis of the barriers encountered by women when they file a complaint and the possible measures taken to overcome them [16, 62–66]. Nevertheless, in the present study we will explore the source of these complaints as a complementary analysis to provide an assessment of this bystander behavior.

## Method

### Design

In order to achieve the intended goal, an empirical ex–post facto study with a descriptive and associative strategy will be used, allowing for the definition, classification and/or categorization of events and, ultimately, a description of the characteristics of the topic of the investigation (which does not require a hypothesis), in addition to an exploration of the functional relationship that exists between variables (particularly between the variables of interest and the variables of sex and age) [67, 68]. Related to this, and based on the previously described results on the topic, it is hypothesized that women will be bystanders in VAW cases and will suggest reporting violence witnessed to a greater extent than men, but no differences are expected to be found according to age.

### Methodological strategy

To reach the intended objective, the methodological strategy used was an analysis of secondary data [67–69]; that is, data that were obtained and processed by other sources, as well as, in this case, quantitative data taken from sociological surveys conducted in Spain by different institutions and administrative data related to the records of complaints for IPVAW.

**Table 1. Information about each sociological survey included.**

| Study | Social perception of gender violence | Social perception of gender violence among adolescents and youth | Macro-survey on violence against women 2014 | Macro-survey on violence against women 2019 |
|---|---|---|---|---|
| | **Study 2968 [43]** | **Study 2992 [49]** | **Study 3027 [40, 41]** | **Study 3235 [42]** |
| **Organism in charge of data collection** | CIS by an agreement established with the GDGV | CIS by an agreement established with the GDGV | CIS by an agreement established with the GDGV | CIS by an agreement established with the GDGV |
| **Population of interest** | Resident in Spain aged 18 and older and | Resident in Spain aged from 15 to 29 years old | Women resident in Spain aged 16 and older | Women resident in Spain aged 16 and older |
| **Type of sample** | Multistage sampling, stratified by clusters with a random proportional selection of the primary sampling units (municipalities) and the secondary units (sections), and a random selection applying quotas for sex and age | Multistage sampling, stratified by clusters with a random proportional selection of the primary sampling units (municipalities) and the secondary units (sections), and a random selection applying quotas for sex and age | Multistage sampling, stratified by clusters with a random proportional selection of the primary sampling units (municipalities) and the secondary units (sections), and a random selection applying quotas for age and occupation | Multistage sampling, stratified by clusters with a random proportional selection of the primary sampling units (municipalities) and the secondary units (sections), and a random selection applying quotas for age and occupation |
| **Data collection dates** | From 19 November to 3 December 2012 | From 17 June to 4 July 2013. | From 19 September to 14 November 2014 | From 12 September to 1st December 2019 |
| **Language of administration** | Spanish | Spanish | Spanish | Spanish |
| **Mode of data collection** | Face-to-face interviews during a home visit | Face-to-face interviews during a home visit | Face-to-face interviews during a home visit | Computer-assisted personal interviews (CAPI) |
| **Projected sample size** | 2,600 people | 2,600 people | 10,258 women | 10,000 women |
| **Achieved sample size** | 2,580 people | 2,457 people | 10,171 women | 9,568 women |
| **Weighting** | No weighting in any case | No weighting in any case | Not applicable in this paper | Not applicable in this paper |
| **Website with technical information and data** | https://analisis.cis.es/cisdb.jsp?ESTUDIO=2968 | https://analisis.cis.es/cisdb.jsp?ESTUDIO=2992 | https://analisis.cis.es/cisdb.jsp?ESTUDIO=3027 | https://analisis.cis.es/cisdb.jsp?ESTUDIO=3235 |

Source: Own elaboration based on technical datasheets available in CIS website.

CIS: Centro de Investigaciones Sociológicas.

GDGV: Government Delegation for Gender Violence.

## Data sources

Data from the sociological surveys described in Table 1 were analyzed.

The microdata and questionnaires are available on the CIS website (see Table 1). All data are previously processed by the CIS and the verification, purging, anonymizing, cataloguing and computer data loading have been completed by the CIS.

The administrative data related to the number of complaints of IPVAW by year and by source of the complaint are fully available in the Statistics Portal of the Government Delegation for Gender Violence [61] since 2009. These data are fully anonymized.

## Variables

Table 2 presents the variables analyzed in each sociological survey, including the questions and possible response alternatives in each case.

Regarding the reports or complaints, the number and source were analyzed using the data provided by the Statistics Portal of the Government Delegation for Gender Violence (governmental website) [61]. It should be noted that with respect to the source of the complaint, available reports in Spain distinguish between:

- Complaints filed directly by the victim or her family before the court or to the police.

- Complaints resulting from police reports, either by a complaint made by the victim, by family members or by direct police intervention.

- Injury reports received directly in the courts.

- Complaints received from support services or third parties in general.

## Data analysis

Regarding the surveys, the CIS was asked to provide the files with their corresponding micro-data (requests can be made directly through their website and are accessible to all researchers). After reviewing the database and determining which variables to use, a descriptive analysis of

**Table 2. Variables analyzed in each sociological survey included (from the questionnaires available in CIS website).**

| Study | Variable | Question | Answers |
|---|---|---|---|
| **Social perception of gender violence Study 2968 [43] Social perception of gender violence among adolescents and youth Study 2992 [49] (same questionnaire in both studies)** | Knowledge of IPVAW cases within the social circle of the participants | 13. Are you aware of any cases of women in your inner circle who are currently or have been previously victims of abuse by their husband/partner or ex-husband/ex-partner? (Question to the whole sample) | . Yes . No . I don't know . No answer |
| | The type of (hypothetical) helping behavior in view of this type of situation | 14. Were you to witness or become aware of a situation involving male aggression or abuse against a woman, what do you think you would do? (Question to the whole sample) | . Nothing . Confronting the offender . Call the police . Attract the attention of other bystanders who might be able to help . I don't know . No answer |
| **Macro-survey on violence against women 2014 Study 3027 [40, 41]** | Report (or not) the IPVAW experienced by the woman interviewed | 38. Did the police have any knowledge of the incidents caused by any of your current or previous partners? (Question to the whole sample) | . Yes . No . No answer |
| | If reported, who filed the report or complaint | 38b. Did you personally inform the police or did another person report the incident? (Question for those who answered yes in question 38) | . You reported . Reported by someone else . No answer |
| | Explain (or not) the IPVAW experienced | 46. I will now mention several different people. Could you tell me whether you told this person about your partner or ex-partner's behavior? (The interviewee may reply yes or not for each case) (Question only for women who had been victim of IPVAW) | . To her mother . To her father . To her daughter . To another female family member . To another male family member . To a female family member of your partner or ex-partner. . To a male family member of your partner or ex-partner . To a female friend . To a female neighbor or work colleague . To a female teacher or tutor . To another people |
| | If explained, reaction of the person to whom the experience was revealed | 46a. How did this person react? (This question is asked for each of the persons to whom the interviewee told the facts) (Question for those who answered yes in question 46) | . Advised you to leave the relationship . Advised you to give him another chance . Reprimanded you for your attitude . Reacted with indifference to the situation . Other answer . No answer |

*(Continued)*

**Table 2.** (Continued)

| Study | Variable | Question | Answers |
|---|---|---|---|
| **Macro-survey on violence against women 2019 Study 3235 [42]** | Report (or not) the IPVAW experienced by the woman interviewed If reported, who filed the report or complaint | M1P16 (current partner) M2P16 (former partner) Thinking back on all these episodes that You have suffered by your **current partner / former partner**, have the Police or Guardia have the police or the Guardia Civil been aware of any of them? (Question only for women who had been victim of IPVAW) | . Yes . No . No answer |
| | | M1P16c (current partner) M2P16c (former partner) Did you report it yourself to the police or did someone else report it? (Question for those who answered yes in question 16) | . You reported . Reported by someone else . No answer |
| | | M1P17 (current partner) M2P17 (former partner) Have you or any other person or institution gone to court to file a complaint? (Question only for women who had been victim of IPVAW) | . Yes, I reported personally . Yes, other person or institution go to the court to file a complaint . No, and none other person or institution go to the court to file a complaint . No answer |
| | Explain (or not) the IPVAW experienced | M1P23 (current partner) M2P23 (former partner) I will now mention several different people. Could you tell me whether you told this person about the behavior of your partner or former partner. . .? (The interviewee may reply yes or not for each case) (Question only for women who had been victim of IPVAW) | . To her mother . To her father . To her daughter . To her brother . To another female family member . To another male family member . To a female friend . To a male friend . To a female neighbor or work colleague . To a male neighbor or work colleague . To a female teacher or tutor . To a male teacher or tutor . To another man . To another female . To other service or organization . None . No answer |
| | If explained, reaction of the person to whom the experience was revealed | M1P23A (current partner) M2P23A (former partner) How did this person react? (This question is asked for each of the persons to whom the interviewee told the facts) (Question for those who answered yes in question M1P23 or M2P23) | . Advised you to leave the relationship . Advised you to give him another chance . Reprimanded you for your attitude . Reacted with indifference to the situation . Other answer . No answer |

Source: Own elaboration based on questionnaires available in CIS website.

the variables (frequency, percentages, bar charts, etc.) was applied, which permitted us to obtain information from the different indicators. It should be noted that the response options "don't know" or "no answer" were recodified as a single category in order to minimize problems with the expected minimum frequencies. Additionally, contingency tables were developed, and comparisons were made using a chi-square analysis to determine the relationship

between the variables studied and sex or age. The SPSS 28.0 (IBM Corporation, Armonk, NY, USA) statistical program was used for data analysis.

In the case of information relative to complaints filed, we present all available data, but only two periods of time were compared: between 2009 (which marks the first year the source used was made available) and 2019; and between 2019 and 2020. In the first case, the reason was to have a comparison over a period of time, and in the second case, it was to differentiate the data corresponding to 2020 and be able to take into account the possible influence stemming from the effect of the Covid-19 pandemic and the subsequent confinement it gave rise to [60–62].

## Results

### Social perception of IPVAW

We begin with the data obtained from the surveys, specifically those related to social perception of IPVAW in the general population [43] and among young people [49].

In the general population, data obtained from the survey conducted in 2012 [43] relative to awareness of cases of IPVAW show that approximately 30% of the subjects interviewed had knowledge of at least some case. This proportion was significantly higher among women (34.8%) relative to men (25.3%) (Chi-square (2) = 27.831, p < .001, Cramer's V = .104), and among participants aged 18 to 49 years (36.7% among those between the ages of 18 and 29, 35.2% among those between the ages of 30 and 39, and 34.4% among those aged between 40 and 49, respectively) relative to those aged 64 and older (17.4%) (Chi-square (8) = 66.678, p < .001, Cramer's V = .153).

An analysis of knowledge according to sex and age (see Fig 1) indicates that, for men and women alike, the knowledge of these type of cases decreases by more than half with age (from slightly greater than 45% to nearly 20% among women, and from slightly more than 30% to nearly 13% among men). The analysis performed specifically indicates a significant relationship between age and knowledge of cases among women as well as among men such that: women aged 18 to 39 report significantly greater knowledge and those aged 64 and older report significantly less knowledge (Chi-square (8) = 47.092, p < .001); and among men, those aged 64 and older have significantly less knowledge of cases of IPVAW than the other age groups (Chi-square (8) = 34.123, p < .001, Cramer's V = .141).

Regarding the hypothetical actions taken by women and men if they were to witness a situation of IPVAW (see Fig 2), the fact that a vast majority consider themselves to be "active" bystanders is noteworthy. In fact, only approximately 2% state that they would do nothing and approximately 5% do not respond or would not know what to do. Notifying the police is the most common response among nearly 60% of men and women regarding their anticipated response. Beyond this general agreement, there is a significant relationship between sex and hypothetical helping behavior (Chi-square (4) = 111.227, p < .001, Cramer's V = .213) whereby women claim to a greater extent that they would call the police or attract the attention of other bystanders who might be able to help, whereas men claim to a greater extent that they would confront the aggressor directly.

By age groups (see Fig 3), similar results were obtained in that the majority of the people in each age group self-identify as active bystanders, and the preferred option among all respondents was to call the police. However, once again, and beyond this general agreement, a significant relationship can be observed between age and hypothetical helping behavior (Chi-square (16) = 54.859, p < .001, Cramer's V = .080) such that the younger respondents were those most likely to choose to confront the aggressor, which is precisely the opposite of what occurs with persons between 50 and 64 years of age, who would to a larger extent call the police and to a lesser extent confront the aggressor. However, those in this older age group would, in

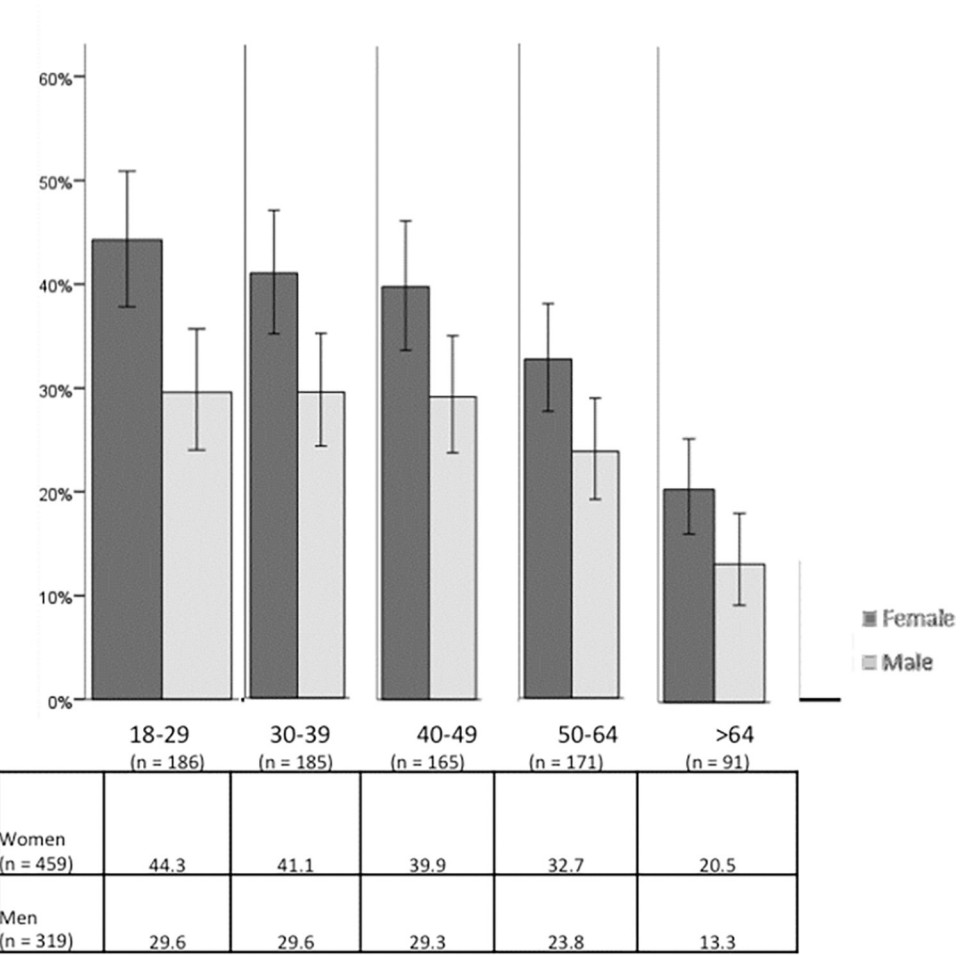

**Fig 1. Percentage of people who know about a case of IPVAW occurring within their social circle, by sex and age.**
Source: Own elaboration based on CIS study 2968 [43], question 13. 95% confidence intervals are provided for each subsample.

turn, be significantly more likely to ask other people for help and less likely to confront the aggressor. Moreover, it should be noted that those from the older age group would consider themselves passive bystanders to a larger extent (nearly 5%).

Combining sex and age variables, a significant relationship can be observed between age and hypothetical response among both men (Chi-square (16) = 31.735, p = .022, Cramer's V = .091) and women (Chi-square (16) = 44.790, p < .001, Cramer's V = .100). In the case of men (see Fig 4), there is a consensus among all age groups with the most selected response being to call the police, followed by confronting the aggressor, although respondents from the older age group were significantly more likely to call the police and significantly less likely to confront the aggressor, while the opposite holds true for the younger age group. Additionally, attracting the attention of other bystanders was by far the most selected response among men between the ages of 30 and 39 compared to the other age groups.

In the case of women (see Fig 5) the choice of hypothetical helping behavior is very similar. There is a consensus among all age groups with the most selected response being to call the police, although women between the ages of 50 and 64 choose this option significantly more

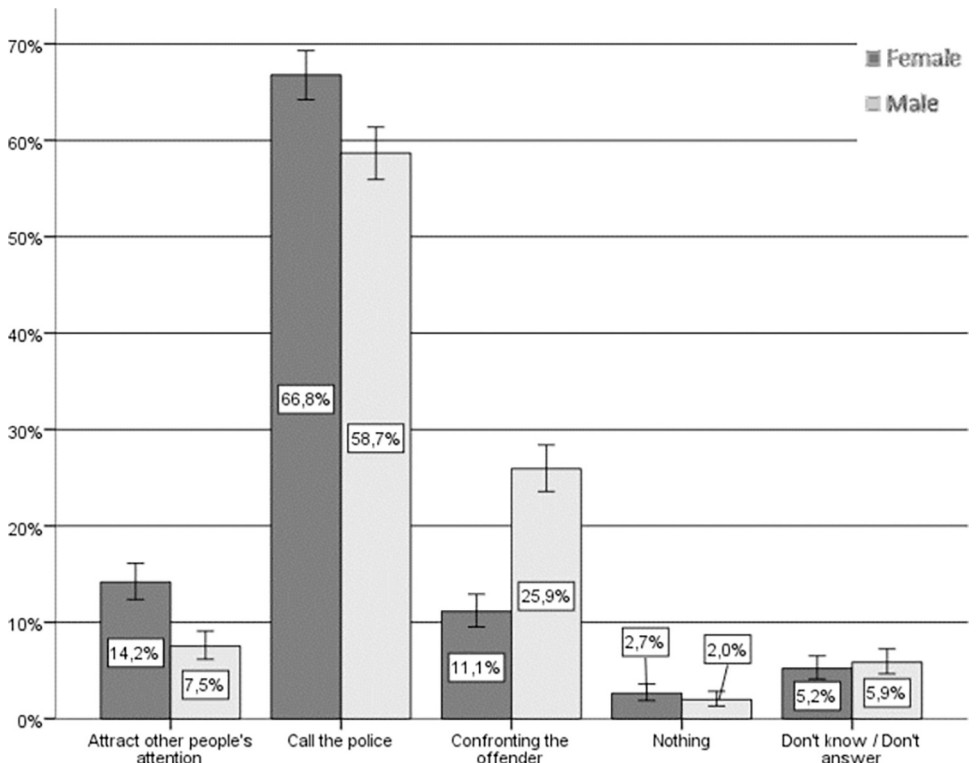

**Fig 2. Hypothetical response to IPVAW, by sex.** Source: Own elaboration based on CIS study 2968 [43], question 14. 95% confidence intervals are provided for each subsample.

while those older than 64 years of age choose this option significantly less. Additionally, attracting the attention of other bystanders is significantly less chosen as a response among women 50 to 64 years of age, while confronting the aggressor is chosen significantly more among women 30 to 39 years of age. Finally, it should be noted that older women self-identify to be more passive bystanders to a greater extent (nearly 6.5%).

Among the younger population, data obtained from the survey conducted in 2013 [49] relative to knowledge of cases of IPVAW seem to be similar to those of the general population. Once again, nearly 30% of the people interviewed are aware of at least one case, with this proportion being significantly greater among women (34.1%) than among men (23.7%) (Chi-square (2) = 32.368, p < .001, Cramer's V = .114). However, in this case, unlike that of the general population, this awareness increases with age and is substantially greater among older people (22.4% in the 15–17 age group and 30.2% in the 18–29 age group) (Chi-square (2) = 10.867, p = .004, Cramer's V = .065).

An analysis of this awareness by sex and age indicates that this increase occurs just as much among men (from 19.% to 24.7%) as among women (from 26.0% to 35.7%), although in the case of women there is a significant relationship between both variables (Chi-square (2) = 7.806, p = .040, Cramer's V = .095).

Regarding the hypothetical actions taken by young women and men if they were to witness a situation of IPVAW (see Fig 6), a vast majority of those interviewed also self-identify as "active" bystanders. Fewer than 2% claim that they would take no action, and approximately 4% do not respond or would not know what to do. Once again, contacting the police is the most common anticipated reaction (approximately 58% of respondents). Likewise, there is

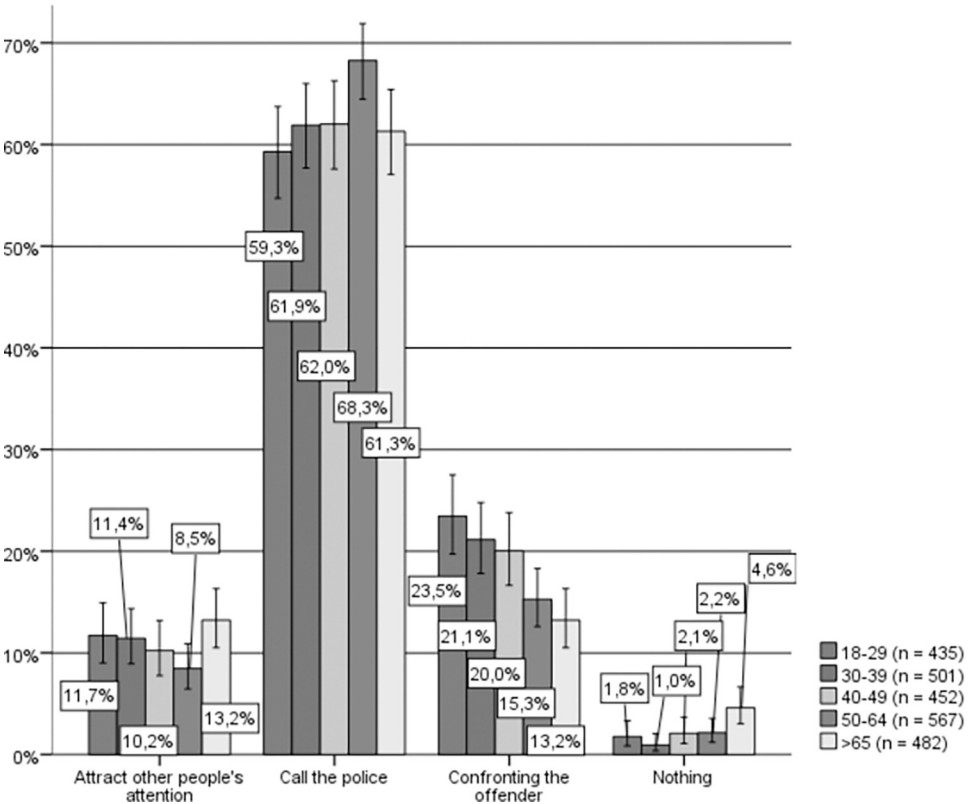

**Fig 3. Hypothetical response to IPVAW, by age groups.** Source: Own elaboration based on CIS study 2968 [43], question 14. 95% confidence intervals are provided for each subsample.

also a significant relationship between sex and hypothetical response (Chi-square (4) = 147.101, p < .001, Cramer's V = .246) which follows the same tendency of the general population; that is, younger women claim to a greater extent that they would contact the police or attract the attention of other bystanders who might be able to help, while younger men claim to a greater extent that they would confront the aggressor.

By age groups (see Fig 7), in both cases the majority of young people self-identify as active bystanders and their preferred response is to call the police, followed by confronting the aggressor, and attracting the attention of other bystanders who might be able to help. However, while a significant relationship between age and hypothetical response can be observed (Chi-square (4) = 12.571, p = .014, Cramer's V = .072), only in the case of attracting the attention of other people was there a significant difference, whereby the younger respondents preferred this option significantly more than the other age groups.

Combining sex and age variables, a significant relationship between age and hypothetical response can be observed only in the case of women (Chi-square (4) = 12.244, p = .032, Cramer's V = .090). Among men (see Fig 8), the consensus between both age groups is that the most selected response is to call the police, followed by confronting the aggressor, and attracting the attention of other bystanders.

In the case of women (see Fig 9), there is also a consensus between both age groups in that the most selected response is to call the police, followed by attracting the attention of other bystanders, although the option of confronting the aggressor is selected significantly more often among young women aged 18 to 29, compared to those aged 15 to 17.

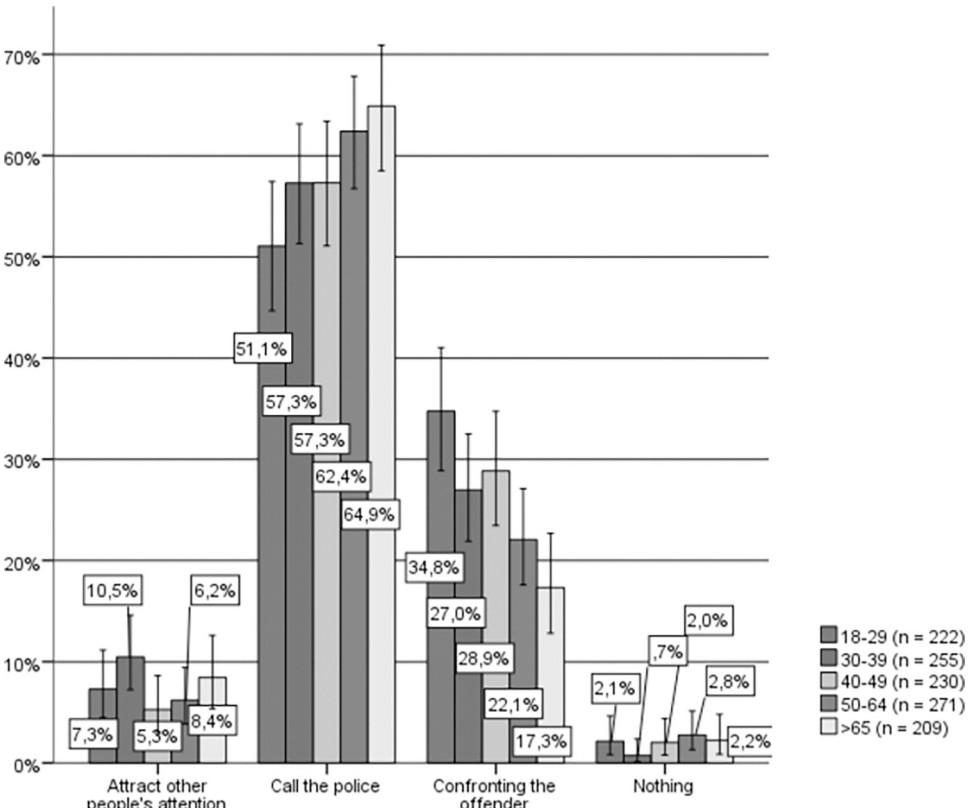

**Fig 4. Hypothetical response to IPVAW, men by age group.** Source: Own elaboration based on CIS study 2968 [43] question 14. 95% confidence intervals are provided for each subsample.

### Perception of victims of IPVAW

Next, data obtained from the macro-surveys conducted in 2014 [40, 41] and 2019 [42] will be presented. We will analize whether the acts of violence were reported and if so, who reported them; and also whether the women had told of her experience to another person and if so, how that person had reacted.

Regarding the macro-survey conducted in 2014 [40, 41], among the women interviewed who had been in an intimate partner relationship at some point in their life (n = 9,807), 16.1% (n = 1,579) had been victim of physical violence and/or sexual violence or had been afraid of their partner or ex-partner. Approximately 1.7% of the women who had experienced this type of situation filed a complaint directly with the courts (n = 28); and the police were informed in 26.9% of the cases (n = 424) (Study 3027, question 38, see Table 2). Among those, the victim herself filed the complaint in 78% of the cases, while in 20.1% of the cases the complaint was filed by another person (1.9% did not respond to this question) (Study 3027, question 38b, see Table 2).

This survey includes a question to determine who knew about the situation of violence and, if so, what their reaction was. According to the available data, 81% of the women who had suffered some form of IPVAW (n = 1,279) had explained their situation to another person, most commonly the women in their inner circle (see Table 3), and particularly to a close female friend, their mother or sister, but also to other female members of their family, neighbors or workmates, and even to female members of the partner or ex-partner's family.

Regarding the reaction of the informed parties with knowledge of IPVAW (see Table 3), in general between 70% and 90% responded as an active bystander by means of providing

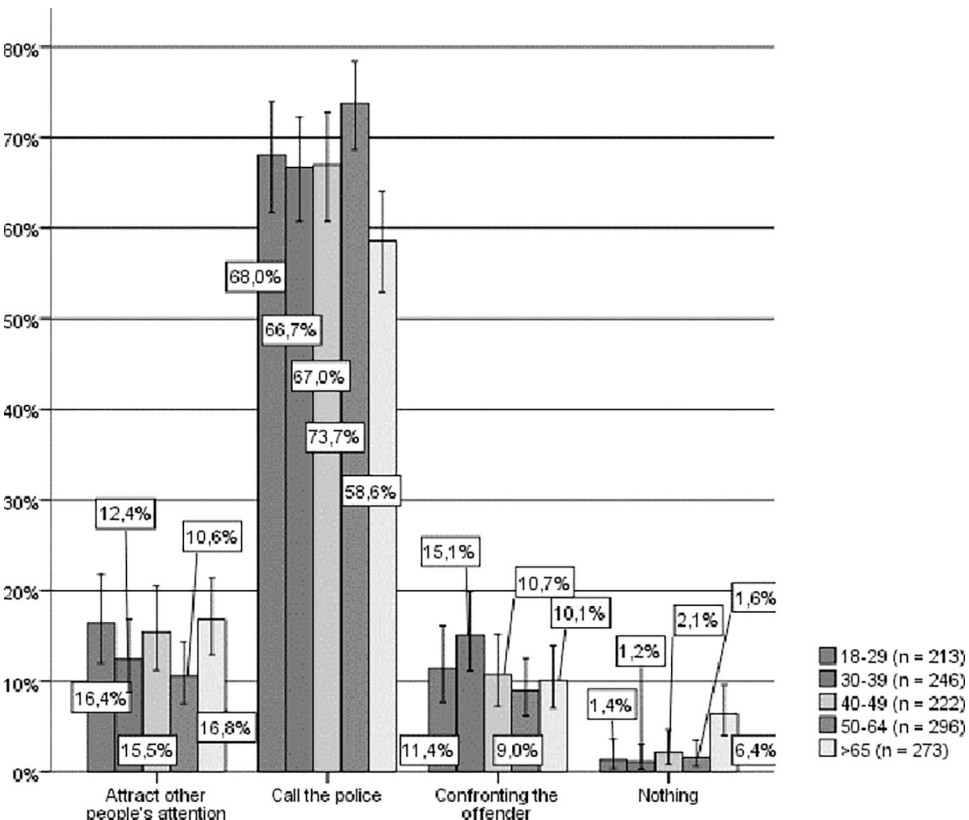

**Fig 5. Hypothetical response to IPVAW, women by age group.** Source: Own elaboration based on CIS study 2968 [43] question 14. 95% confidence intervals are provided for each subsample.

support such as advising the woman to leave the violent relationship. The remainder of the bystanders are classified as those who do not answer, those who provide an uncoded response and those whose response would be considered inactive or passive (showing indifference or encouraging the woman to give the abuser another chance) or even 'negative' (admonishing the victim for her attitude). In this general context, two points stand out: first of all, the fact that slightly more than 13% of mothers advise their daughters to remain with the abuser, which leads us to reflect on the weight of traditional female socialization in our context; and secondly, in terms of a general tendency toward active response, the stark contrast noted among family members of the abuser whereby support was provided in fewer than 50% of the cases, indifference reached 20% and admonishing the victim or suggesting that she stay with the abuser ranges from 13% to 20%.

The macro-survey conducted in 2019 [42] included questions similar to those used in the surveys conducted in the previous wave and, moreover, all information related to the current and former partners is presented separately.

Among the women interviewed who had previously had a partner (n = 9,218), 33.6% (n = 3,098) suffered physical, sexual and/or psychological (emotional, controlling, financial or fear-inducing violence) abuse at the hands of their partner or ex-partner at some point in their life; 14.8% experienced physical and/or sexual (n = 1,362) violence; and 33.2% suffered psychological trauma (n = 3,056). Among women who had a partner at the time of the interview (n = 6,506), 14.7% (n = 958) had suffered physical, sexual and/or psychological violence at the hands of their partner or ex-partner in the 12 months prior to the interview, 2.9% physical

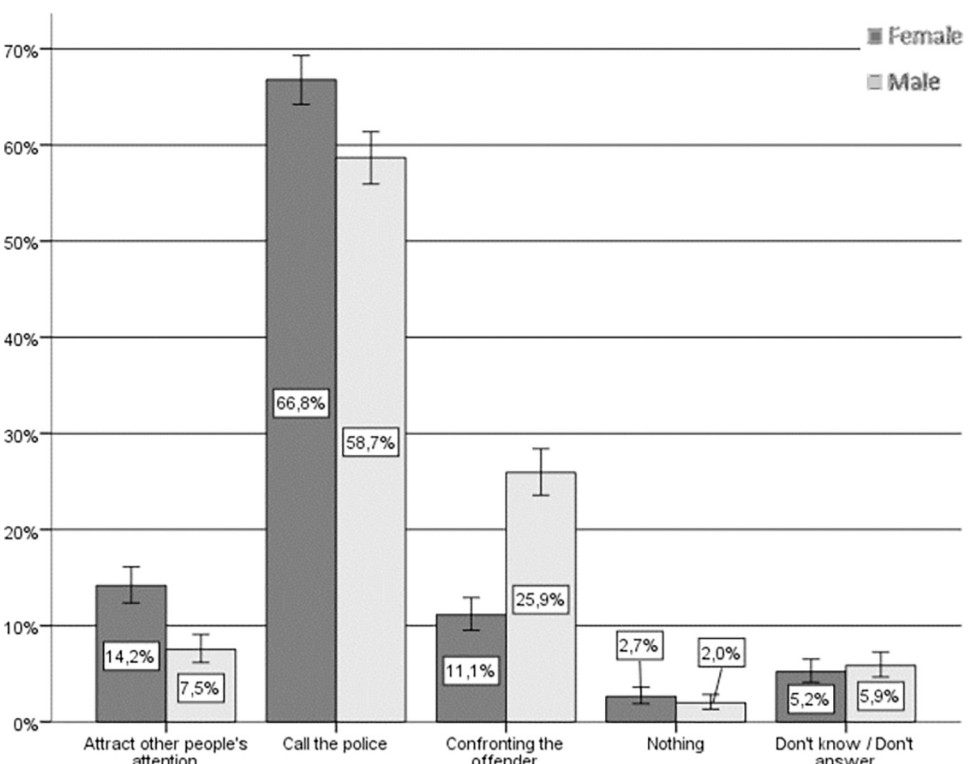

**Fig 6. Hypothetical response to IPVAW among young people by sex.** Source: Own elaboration based on CIS study 2992 [49] question 14. 95% confidence intervals are provided for each subsample.

and/or sexual violence (n = 191), and 14.5% psychological violence (n = 943). It should be noted that questions related to complaints filed for IPVAW and related to knowledge within her inner circle were only used with women who admitted to having experienced physical, sexual or emotional violence or fear at the hands of any partner (n = 2,395) or their current partner (n = 592).

Among all women who had suffered physical, sexual or emotional violence or had feared their partner (n = 2,395), 21.7% (n = 520) had reported these aggressions either to the police or directly to the courts (25% in the case of former partners; 5.4% in the case of current partners) (Study 3235, questions M1P16/M2P16, M1P16c/M2P16c, M1P17/M2P17, see Table 2). One relevant note regarding the methodology is that, although the percentage of reports filed appears to be less than those of the macro-survey in 2014 (28.6%), this question was not posed to those who had suffered emotional violence. Calculating this percentage in the same way as the previous survey (that is, excluding women who had experienced emotional trauma) the figure stands at 28.7%, indicating a lack of change in the rate of filed reports. Moreover, if we consider only those who suffered physical and/or sexual violence, the rate of filed reports stands at 32.1%, indicating a significant difference between reports against a former (34.3%) and current (12.5%) partner.

In the case of violent abuse inflicted by former partners, the police had knowledge of 23.7% of cases (1.3% were reported directly in the courts); in 80.4% of these cases, it was the actual victim who filed the report whereas in 19.5% of the cases it was another person who did so (with 0.1% not responding to the question). In the case of violent abuse inflicted by the current partner, the police had knowledge of 5.4% of the cases (no complaints were filed directly to the courts); in 83.5% of these cases, it was the actual victim who filed the report whereas in 16.5% of the cases it was another person who did so.

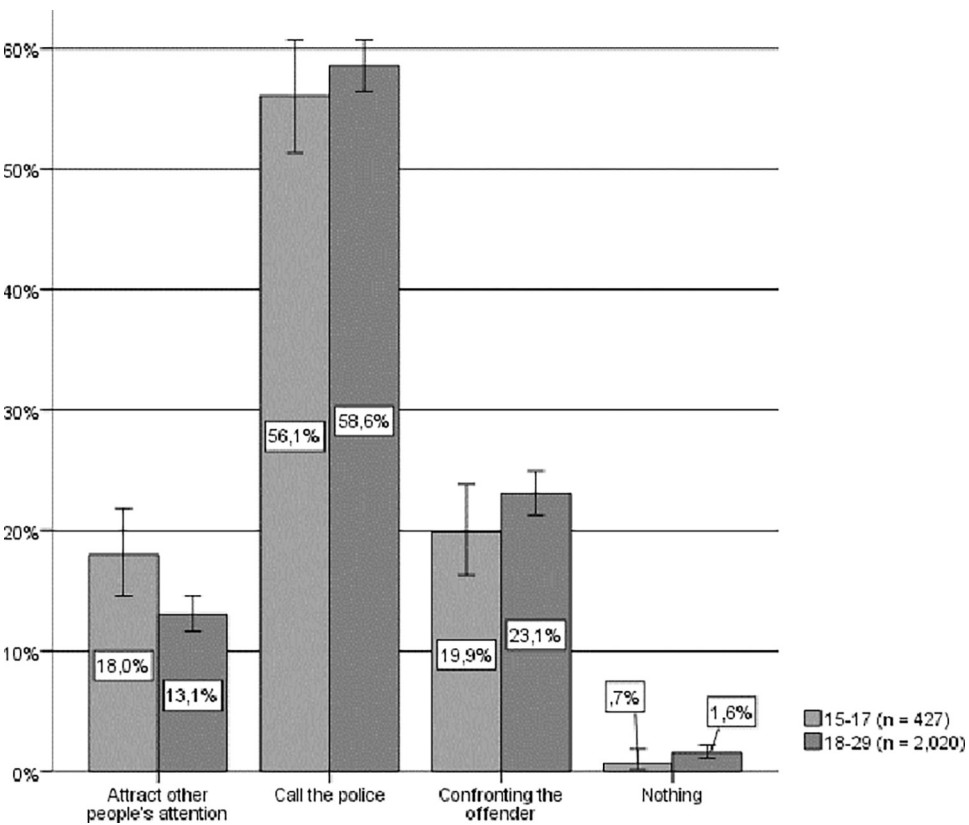

**Fig 7. Hypothetical response to IPVAW of young people, by age group.** Source: Own elaboration based on CIS study 2992 [49], question 14. 95% confidence intervals are provided for each subsample.

The questions used in the survey to determine who knew about the violent situation and, if so, what their reaction was, were worded differently for former partners and current partners. According to the data available from the survey, 77.9% of the women who had suffered some type of violence at the hands of their current or former partner had explained their situation to another person (80.9% of those who had experienced the violence in a previous relationship, and 61.9% of those who were currently in a violent relationship), primarily to women in their inner circle (see Table 4) and more specifically to a female friend, mother or sister, but also to other female members of their family, neighbors or a workmate.

Among the different reactions identified in these surroundings (see Table 4), there is a notable difference between reactions to previous partners or current partners. In the case of previous partners (as was the case in the macro-survey of 2014), the predominant reaction among people comprising the victim's inner circle (greater than 75% in practically all the cases) was to take on the role of active witness by means of offering support such as advising the woman to leave the violent relationship. On the other hand, subscribing to roles that could be considered inactive or passive (displaying outright indifference or urging the woman to give the abuser another chance) or even 'negative' (reprimanding the victim for her attitude) is, in general, quite infrequent if not uncommon (with a higher percentage occurring among cases with a low response rate, such as teachers). We are once again drawn by the fact that those who more frequently advise the women to remain with the abuser are the mothers (8.4%) and other women in the family (7.4%). The indifference and reprimand toward women remains at a low level (below 9% among all groups).

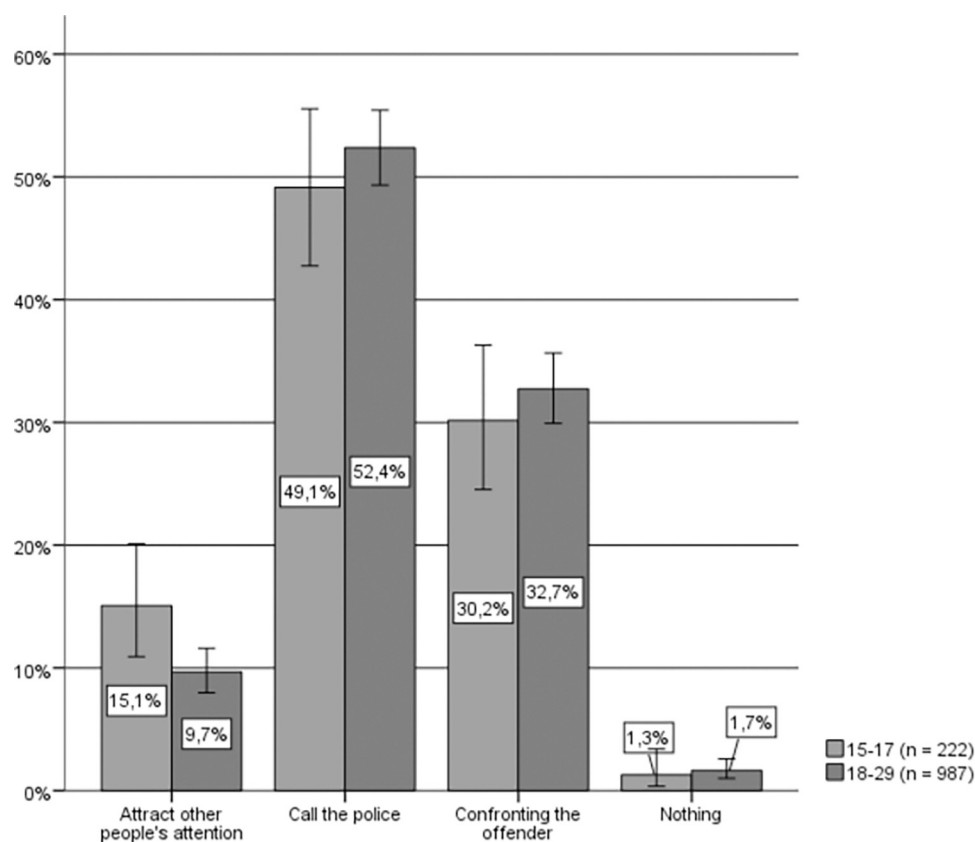

**Fig 8. Hypothetical response to IPVAW, young men by age group.** Source: Own elaboration based on CIS study 2992 [49], question 14. 95% confidence intervals are provided for each subsample.

Conversely, in the case of current partners, although the majority of the groups representing the woman's inner circle continue to take on the role of active witness by means of offering support such as advising the woman to leave the violent relationship, the proportion of those who do so is substantially lower (57.1% in the case of fathers, 56.3% in the case of male friends), and it is not always the main option. For example, 2/3 of the other males in the family opt for giving the abuser another chance (between 17–31% of those in the woman's inner circle choose this option, except in the case of teachers) while 40.0% of brothers and 32.4% of sisters choose another response (uncoded) as their primary option.

## Reporting IPVAW

Finally, regarding the source of the reports filed, the highest proportion (between 75–80%) came from police statements, primarily from reports filed by the victim (between 60–70%) and the rest from direct police intervention, and only a small percentage of reports coming from other sources (Table 5). Complaints filed by assistance services and third parties in general comprised 1–5% of the total, and those filed by family members (either directly or through a police statement) ranged between 1–2.6%.

An analysis of the evolution of IPVAW complaints according to sources between 2009 and 2020 (see Table 5) indicates a significant relationship between these variables (Chi-square (66) = 44240.211, $p < .001$, Cramer's V = .066). Available data show an upward trend in the number of complaints observed in the previous decade, which seems to have been interrupted in

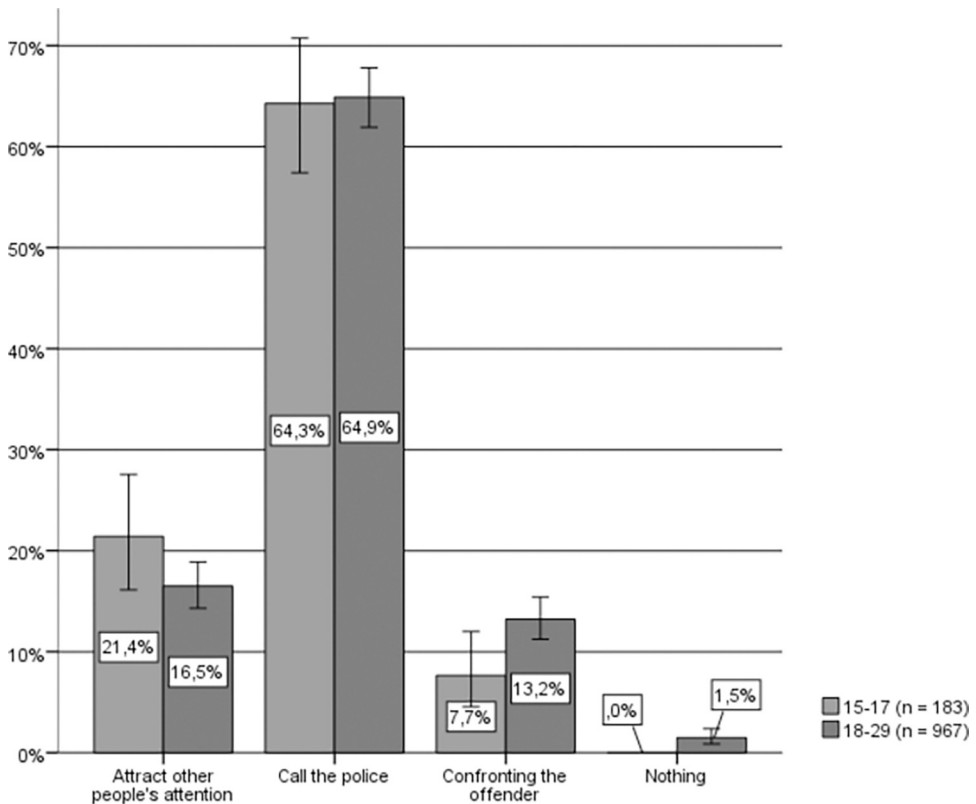

**Fig 9. Hypothetical response to IPVAW, young women by age group.** Source: Own elaboration based on CIS study 2992 [49], question 14. 95% confidence intervals are provided for each subsample.

**Table 3. Knowledge of IPVAW within the victim's inner circle and their reactions.** Macro–survey 2014 (n = 1,279).

| | Knowledge of IPVAW | Advised her to leave the relationship | Advised her to give him another chance | Reprimanded her for her attitude | Reacted with Indifference to the situation | Other answer | No answer |
|---|---|---|---|---|---|---|---|
| **Female friend** | 864 (54.7%) | 731 (84.6%) | 41 (4.7%) | 9 (1.1%) | 18 (2.1%) | 56 (6.4%) | 9 (1.1%) |
| **Mother** | 633 (40.1%) | 456 (72.1%) | 85 (13.4%) | 15 (2.4%) | 26 (4.1%) | 43 (6.8%) | 7 (1.1%) |
| **Daugther** | 509 (32.2%) | 411 (80.8%) | 25 (5.0%) | 11 (2.1%) | 20 (4.0%) | 35 (6.9%) | 6 (1.2%) |
| **Father** | 316 (20.0%) | 245 (77.4%) | 21 (6.6%) | 13 (4.1%) | 15 (4.9%) | 19 (6.0%) | 3 (0.9%) |
| **Female family member** | 307 (19.5%) | 242 (78.8%) | 17 (5.5%) | 9 (2.9%) | 10 (3.4%) | 26 (8.5%) | 3 (0.8%) |
| **Female neighbour or work colleague** | 238 (15.1%) | 211 (88.6%) | 6 (2.6%) | 1 (0.4%) | 6 (2.7%) | 14 (5.8%) | - |
| **Male family member** | 271 (17.2%) | 215 (79.3%) | 7 (2.6%) | 3 (1.1%) | 17 (6.3%) | 27 (9.9%) | 2 (0.7%) |
| **Female family member of your partner/ex partner** | 265 (16.8%) | 103 (38.8%) | 53 (19.9%) | 35 (13.2%) | 50 (18.8%) | 22 (8.2%) | 3 (1.2%) |
| **Male family member of your partner/ex partner** | 139 (8.8%) | 60 (42.9%) | 18 (12.6%) | 23 (16.2%) | 28 (19.8%) | 10 (6.8%) | 2 (1.6%) |
| **Other people** | 87 (5.5%) | 73 (83.2%) | 1 (1.2%) | 4 (4.8%) | 6 (7.2%) | 6 7.2%) | 3 (3.7%) |
| **Female teacher or tutor** | 33 (2.1%) | 24 (74.1%) | 1 (3.3%) | 3 (8.9%) | 2 (7.0%) | 2 (7.0%) | 2 (7.0%) |

Source: Own elaboration based on data obtained from the macro-survey conducted in 2014 [40, 41], questions 46 and 46a.

**Table 4. Knowledge of IPVAW within the victim's inner circle and their reactions.** Macrosurvey 2019.

| IPVAW any partner n = 2,395<br>IPVAW current partner n = 592 | | Knowledge of IPVAW | Advised her to leave the relationship | Advised her to give him another chance | Reprimanded her for her attitude | Reacted with Indifference to the situation | Other answer | No answer |
|---|---|---|---|---|---|---|---|---|
| **Female friend** | Any partner | 1069 (54.7%) | 908 (84.9%) | 38 (3.6%) | 4 (0.4%) | 14 (1.3%) | 95 (8.9%) | 11 (1.0%) |
| | Current | 194 (32.8%) | 65 (33.2%) | 43 (21.9%) | 4 (2.0%) | 15 (7.7%) | 60 (30.6%) | 9 (4.6%) |
| **Mother** | Any partner | 775 (39.7%) | 584 (75.4%) | 65 (8.4%) | 34 (4.4%) | 16 (2.1%) | 73 (9.4%) | 3 (0.4%) |
| | Current | 111 (18.8%) | 41 (37.3%) | 31 (28.2%) | - | 13 (11.8%) | 25 (22.7%) | - |
| **Daugther** | Any partner | 511 (26.2%) | 410 (80.4%) | 31 (6.1%) | 5 (1.0%) | 13 (2.5%) | 47 (9.2%) | 4 (0.8%) |
| | Current | 110 (18.6%) | 30 (27.0%) | 23 (20.7%) | 2 (1.8%) | 20 (18.0%) | 36 (32.4%) | - |
| **Father** | Any partner | 416 (21.3%) | 334 (80.3%) | 26 (6.3%) | 4 (1.0%) | 14 (3.4%) | 35 (8.4%) | 3 (0.7%) |
| | Current | 35 (5.9%) | 20 (57.1%) | 6 (17.1%) | - | 5 (14.3%) | 4 (11.4%) | - |
| **Brother** | Any partner | 306 (15.7%) | 249 (81.4%) | 19 (6.2%) | 1 (0.3%) | 15 (4.9%) | 22 (7.2%) | - |
| | Current | 35 (5.9%) | 11 (31.4%) | 6 (17.1%) | 1 (2.9%) | 3 (8.6%) | 14 (40.0%) | 0 |
| **Male friend** | Any partner | 288 (14.7%) | 238 (82.6%) | 9 (3.1%) | 4 (1.4%) | 6 (2.1%) | 28 (9.7%) | 3 (1.0%) |
| | Current | 32 (5.4%) | 18 (56.3%) | 6 (18.8%) | - | 3 (9.4%) | 5 (15.6%) | - |
| **Another female family member** | Any partner | 244 (12.5%) | 193 (79.4%) | 18 (7.4%) | 1 (0.4%) | 8 (3.3%) | 21 (8.6%) | 2 (0.8%) |
| | Current | 63 (10.6%) | 25 (40.3%) | 17 (27.4%) | 3 (4.8%) | 3 (4.8%) | 13 (21.0%) | 1 (1.6%) |
| **Female neighbour or work colleague** | Any partner | 136 (6.9%) | 104 (77.0%) | 6 (4.4%) | 1 (0.7%) | 6 (4.4%) | 16 (11.9%) | 2 (1.5%) |
| | Current | 18 (3.0%) | 7 (36.8%) | 5 (26.3%) | 1 (5.3%) | 2 (10.5%) | 4 (21.1%) | - |
| **Another male family member** | Any partner | 107 (5.5%) | 94 (92.2%) | 4 (3.9%) | - | 4 (3.9%) | - | - |
| | Current | 32 (5.4%) | 8 (25.0%) | 10 (31.3%) | 3 (9.4%) | 2 (6.3%) | 8 (25.0%) | 1 (3.1%) |
| **Other female** | Any partner | 76 (3.9%) | 65 (85.5%) | 2 (2.6%) | 1 (1.3%) | 2 (2.6%) | 6 (7.9%) | - |
| | Current | 15 (2.5%) | 8 (50.0%) | 4 (25.0%) | - | 2 (12.5%) | - | 2 (12.5%) |
| **Male neighbour or work colleague** | Any partner | 65 (3.3%) | 51 (78.5%) | 2 (3.1%) | - | 4 (6.2%) | 8 (12.3·%) | - |
| | Current | 6 (1.0%) | 3 (60.0%) | 1 (20.0%) | - | 1 (20.0%) | - | - |
| **Other male** | Any partner | 49 (2.5%) | 31 (63.3%) | - | 3 (6.1%) | 4 (8.2%) | 11 (22.4%) | - |
| | Current | 6 (1.0%) | 1 (16.7%) | 2 (33.3%) | - | 1 (16.7%) | 2 (33.3%) | - |
| **Female teacher or tutor** | Any partner | 14 (0.7%) | 7 (50.0%) | - | 1 (7.1%) | - | 4 (28.6%) | 2 (14.3%) |
| | Current | 1 (0.2%) | 1 (100%) | - | - | - | - | - |
| **Male teacher or tutor** | Any partner | 10 (0.5%) | 10 (100%) | - | - | - | - | - |
| | Current | 1 (0.2%) | 1 (100%) | - | - | - | - | - |

Source: Own elaboration based on data obtained from the macro-survey conducted in 2019 [42], questions M1P23 / M2P23 and M1P23A / M2P23A.

**Table 5. IPVAW reports filed according to source.**

|  |  | 2009 | 2010 | 2011 | 2012 | 2013 | 2014 | 2015 | 2016 | 2017 | 2018 | 2019 | 2020 | Total |
|---|---|---|---|---|---|---|---|---|---|---|---|---|---|---|
| **Victim (directly court)** | n | 10,871 | 11,158 | 12082 | 10,750 | 12,269 | 9,769 | 5,238 | 4,607 | 5,990 | 4,676 | 3,928 | 2,659 | 93,997 |
|  | % | 8.0% | 8.3% | 9.0% | 8.4% | 8.9% | 7.7% | 4.1% | 3.2% | 3.6% | 2.8% | 2.3% | 1.8% | 5.5% |
|  | A SR | 42.4 | 47.2 | 58.8 | 46.9 | 69.6 | 35.8 | -23.7 | -39.8 | -35.7 | -50.9 | -60.0 | -66.7 |  |
| **Family (directly)** | n | 451 | 487 | 450 | 435 | 625 | 651 | 1,504 | 375 | 444 | 768 | 956 | 246 | 7,392 |
|  | % | 0.3% | 0.4% | 0.3% | 0.3% | 0.5% | 0.5% | 1.2% | 0.3% | 0.3% | 0.5% | 0.6% | 0.2% | 0.4% |
|  | A SR | -5.8 | -4.0 | -5.6 | -5.3 | 3.8 | 4.6 | 41.7 | -10.3 | -10.8 | 1.8 | 8.9 | -16.7 |  |
| **Victim (report police)** | n | 87,635 | 86,760 | 83,693 | 81,836 | 75,767 | 78,758 | 83,667 | 94,802 | 108,945 | 110,623 | 116,990 | 105,087 | 1,114,563 |
|  | % | 64.7% | 64.7% | 62.5% | 63.7% | 60.7% | 62.1% | 64.8% | 66.0% | 65.5% | 66.3% | 69.6% | 69.7% | 65.2% |
|  | A SR | -4.6 | -4.3 | -22.2 | –12.0 | -35.2 | -24.0 | -3.7 | 6.8 | 2.7 | 9.3 | 39.3 | 38.1 |  |
| **Family (report police)** | n | 1,436 | 1,697 | 1,092 | 1,189 | 1,247 | 1,421 | 1,595 | 1,685 | 2,957 | 2,716 | 3,406 | 2,840 | 23,281 |
|  | % | 1.1% | 1.3% | 0.8% | 0.9% | 1.0% | 1.1% | 1.2% | 1.2% | 1.8% | 1.6% | 2.0% | 1.9% | 1.4% |
|  | A SR | -10.0 | -3.2 | -18.0 | -14.1 | -11.5 | -7.7 | -4.1 | -6.4 | 15.4 | 9.8 | 24.7 | 18.3 |  |
| **Police (directly)** | n | 17,445 | 18,137 | 19,633 | 17358 | 18,222 | 18,984 | 20,131 | 23,635 | 25,600 | 25,006 | 22,084 | 20,265 | 246,500 |
|  | % | 12.9% | 13.5% | 14.7% | 13.5% | 14.6% | 15.0% | 15.6% | 16.5% | 15.4% | 15.0% | 13.1% | 13.4% | 14.4% |
|  | ASR | -17.0 | -9.8 | 2.4 | -9.7 | 1.7 | 5.8 | 12.3 | 23.0 | 11.9 | 6.7 | -15.9 | -11.4 |  |
| **Third party** | n | 1,563 | 1,226 | 1,762 | 2,182 | 2,400 | 2,130 | 2,483 | 3,920 | 6,132 | 7,921 | 4,696 | 5,463 | 41,878 |
|  | % | 1.2% | 0.9% | 1.3% | 1.7% | 1.9% | 1.7% | 1.9% | 2.7% | 3.7% | 4.7% | 2.8% | 3.6% | 2.5% |
|  | A SR | -32.2 | -37.9 | -28.0 | -18.1 | 12.6 | -18.4 | -12.8 | 7.2 | 34.3 | 63.8 | 9.5 | 30.8 |  |
| **Injury report in court** | n | 16,138 | 14,640 | 15,290 | 14,727 | 14,363 | 15,029 | 14,575 | 14,511 | 16,192 | 15,251 | 16,108 | 14,242 | 181,066 |
|  | % | 11.9% | 10,9% | 11.4% | 11.5% | 11.5% | 11.9% | 11.3% | 10.1% | 9.7% | 9.1% | 9.6% | 9.5% | 10.6% |
|  | A SR | 16.3 | 4.0 | 10.1 | 10.5 | 10.8 | 15.2 | 8.3 | -6.3 | -12.0 | -20.4 | -14.3 | -15.2 |  |
| **Total** |  | 135,540 | 134,105 | 134,002 | 128,477 | 124,893 | 126,742 | 129,193 | 143,535 | 166,260 | 166,961 | 168,168 | 150,785 | 1,708,677 100% |

ASR: Adjusted Standardized Residuals; Victim (directly): complaints filed directly by the victim before the court or the police; Family (directly): complaints filed directly by the victim family before the court or the police; Victim (report police): complaints resulting from police reports, by a complaint made by the victim; Family (report police): complaints resulting from police reports, by a complaint made by the victim family members.; Police (directly): complaints resulting from police reports by direct police intervention.; Third party: complaints received from support services or third parties in general.; Injury report in court: injuries reports received directly in the courts.

Source: Own elaboration based on data from the Statistics Portal of the Government Delegation for Gender Violence [61].

2020. Police statements from direct police intervention have increased until 2018; in 2019 there was a decrease and in 2020 there seems to be a slight increase. A similar trend was found regarding complaints filed by support services or third parties, which have increased considerably between 2016 and 2018, and both a significant decrease in 2019 and an increase in 2020, although the numbers are still relatively small.

The complaints filed directly in the courts by the victim and the reports of injuries received directly by the courts have decreased significantly from 2015 on, and in 2020 direct complaints from the victim have decreased even more. The opposite trend is observed regarding police statements filed by the victim, with these being the predominant source of complaints over the last decade (about 65–70% of the cases); police reports from victims have increased considerably from 2015 on, and even more from 2019.

Finally, the number of police reports filed by family members and complaints filed directly by family members has also increased, although to a lesser degree, and the number they represent is also small. However, a significant decrease was observed in both cases in 2020, especially in those complaints presented directly, which dropped to the lowest proportion of the series (0.2%).

## Discussion

A review of the different secondary data sources used has enabled us to reach our proposed objective, obtaining relevant information to estimate the volume of bystanders in cases of IPVAW in Spain, and the type of response and helping behavior they exhibit.

First of all, the surveys on the perception of IPVAW [43, 49] conducted between 2012–2013 allow us to conclude that about 1/3 of the population, including both the general population and the younger population, claim to be aware of some type of this violence, especially female friends and neighbors. It merits to point out that this level of knowledge is significantly greater than what was found in other surveys. For example, the first two waves of so-called macro-surveys conducted in Spain [70–72] indicated that slightly more than 4% of the women interviewed in 1999 and 2002 were aware of a woman in their immediate circle of friends or family who was a victim of some sort of IPVAW; and the subsequent two macro-surveys waves [38, 39, 73] indicated that approximately 18% of those interviewed in 2006 and 2011 were aware that within their inner family circle there was a female victim of IPVAW, especially female friends or neighbors. The Euro-barometer on the subject matter [27, 28] indicated that 23% - 18% of those interviewed knew a victim of IPVAW within their inner circle of friends or family, 21%-14% in their neighborhood, and 7%-6% in their place of study or work, respectively. Likewise, the survey on the perception of VAW conducted in the Basque Country [44] indicated that only about 10% of those interviewed had knowledge of a case of IPVAW. In contrast, recent studies, such as León et al. [12], indicate a considerably higher level of knowledge of cases, nearly 46.2% of participants, suggesting that the greatest amount of information and awareness on the matter could be positively influenced by the knowledge and acknowledgement of these cases on behalf of the general public. In short, although survey data used in this study have some limitations (i.e., in some surveys only a dichotomous question is used, so a person answering positively could be aware of one or multiple cases of IPVAW; or the macro-surveys only account for cases in which the victims actively reported the cases to other people, but fail to account for other sources by which people could be aware of such cases, thus underrepresenting actual awareness of this violence), the results obtained and their comparison with the previous and recent research suggest that IPVAW awareness is increasing quickly in Spain.

An analysis by sex and age of the information contained in the surveys on perception of IPVAW [43, 49] showed that awareness of cases of this type of violence is nearly 10 points higher among women. Regarding age, although awareness among women was higher in all age groups, those in the younger (15–17 years) and older (64 and older) age groups had greater awareness, although the highest figure was seen in the 18 to 29 age group, after which the numbers gradually decreased. These same effects of sex and age were also noted in the survey conducted in the Basque Country [44]. In respect of these effects, women having greater awareness is easily explained by the fact that victims tend to confide in other women about the violence they have experienced, as shown in the results of our study. In cases of IPVAW, attitudes of rejection toward the aggressors and support for the victims, existing to a greater extent among women (i.e., Sanchez-Prada et al. [74]), help explain the preference among victims to confide in women from their inner circle more so than men. In respect of age, the data obtained underscore the existence of a certain "age effect" (with a greater degree of awareness among those 18–20 years of age, and a subsequent decline), similar to the effect identify in other research (i.e., Waterman et al. [59]), which could be related to awareness campaigns on the issue of this type of violence that have been carried out over the last two decades [75]. This may be added to the effect of awareness of cases among their peers and attitudes of rejection toward IPVAW and support toward victims held in large part by persons in that age group [74, 76]. These results are particularly important because they give us an indication of the

target population towards which campaigns to increase IPVAW case recognition should be directed. In particular, these results point to the need to work on turning men and older people into active witnesses of violence who can collaborate in its prevention and eradication, without neglecting the younger population.

Regarding the type of helping behavior informed by those with knowledge of cases of IPVAW (hypothetical response), the data taken from the surveys on the perception of this type of violence [43, 49] indicate that the overwhelming majority (approximately 90%) self-identify as active bystanders, calling the police being the most commonly selected response among both men and women in all age groups. A comparison between sexes and age reveals this as by far the most preferred option among women compared to men overall, in the 50 to 64 age group and women of that age, and among men 64 years of age and older. The option of confronting the aggressor is the second most chosen option among young men and, in some age groups among women (the 18–29 age group among young women and the 30–39 age group among general population women). Taking an approach of passive bystander appears to be less common in general and is relevant only among older women.

It is worth noting that these results are, overall, similar to those obtained in the survey conducted in the Basque Country [44], in which the majority of respondents indicated that they would respond as an active bystander, calling the police being the preferred response (rather than helping the IPVAW victim); those who considered themselves to be passive bystanders associated intervention with a perceived negative consequence, with a perception of inability, or with an absence of sense of responsibility, as described in the literature on this subject [8–10, 14–16, 20, 21]; and by sex a similar result was also found, although the number of passive bystanders was higher among younger age groups. On the other hand, these results differed from those obtained in other studies [10, 18, 77] in that bystanders preferred other responses (such as offering help or support, helping the victim decide what to do, or even talking directly with the partner), and reserved the option of calling the police for cases of more extreme violence. These differences may be highly dependent on the responses provided in the surveys analyzed because, while other researches offer a wider variety of responses, potentially leading to more selections, these surveys only included three active responses. Ultimately, as noted by León et al. [12], the number of bystanders who claim willingness to file a formal complaint of IPVAW varies considerably from one study to another, revealing that this willingness may proceed from different factors (such as the personal characteristics of the bystander or the incidents of violence), which would require a more in-depth analysis. This study in particular corroborates the existence of a relationship between the type of behavioral response in which a bystander engages and personal variables of the bystander such as sex and age, as similarly described in previous studies on the subject [60, 77, 78].

An analysis of the information contained in the 2014 [40, 41] and 2019 [42] macro-surveys permits us to deduce that approximately 80% of the women who have experienced IPVAW at some point in their life shared the experience with somebody from their inner circle, especially to a woman (friend, mother, sister, etc.). This result follows in the same line as the results obtained from the survey conducted in Catalonia [45, 46], which showed that 68.5% of female victims of IPVAW had informed either a family member (52.5%), or friend (30.5%) about the situation, and confirms that, as indicated in the literature on the subject [27–30], the majority of women who have experienced this type of violence turn to their inner circle for help. However, in the specific case of current IPVAW, this number falls considerably (up to 20 points) [42], which leads us to think that women may need a certain amount of time to identify and come to terms with the violent situation experienced before being able to explain the situation or request help.

Regarding the response within the victim's inner circle upon becoming aware of a situation of IPVAW, both macro-surveys [40–42] indicate that the most common response is to advise the woman to abandon the violent relationship; that is, bystanders would overwhelmingly take on an active role to support the victim. Only in certain cases (family members of the aggressor, the mother) do other options notably appear, such as giving the abuser another chance. Equally interesting is the important difference between responses of close friends and family to a past and present case of IPVAW. With past cases of IPVAW, not only is the extent of awareness greater within the inner circle, but also, as previously indicated, there is a much stronger tendency toward active response. In contrast, in current or more recent cases of IPVAW, not only is there an observable reduction in the extent of awareness, but there is also a significant increase in the number of bystanders who advise the women to give the abuser another chance (nearly 1/3 of those in the inner circle). This finding suggests the need to delve deeper into the scope and motivation of these responses (ranging from possible anticipation of negative consequences by witnesses, to inaccurate recall or magnification of past active bystanders responses by victims).

It should be noted that although the macro-surveys do not include the option to bystanders of calling the police, they do include a question relative to the filed complaints, from which we can determine that approximately 25% of the women who took part in the macro-survey and were victims of IPVAW filed a report, of which 80% were filed directly by the victim and only 20% were filed by another person. A review of the data for reports filed in Spain offers even fewer results [61], underscoring the fact that for the years studied (2009 to 2020), reports filed by family members of the IPVAW victims are barely 0.4% and 1.4% of the total number.

The difference in the data could suggest that while an analysis of the response among the general population of Spain to surveys on the perception of IPVAW [43, 49] and the response among victims of IPVAW in the macro-surveys [40–42] tends to identify those within the victim's inner circle as active bystanders who would call the police if they were to become aware of this type of violence, the actual records indicate a much less optimistic reality where the levels of involvement among those in the victim's inner circle are low or very low and the intended behavioral response does not actually occur. This could be due to either the self-reporting bias of the social desirability effect for both bystanders and victims [12], a widely documented effect with respect to this type of violence [79, 80], or to the existence of a significant distance between intended and actual behavior, as noted by Azjen and Fishbein's Theory of Planned Action [81], which has also been identified in the literature related to the behavior of bystanders [29, 37]. In our view, highlighting the significant gap between what IPVAW bystanders say they will do and what they actually do (as shown by the data on complaints) and pointing out the need to further delve into the factors that explain theses contradictions are the most relevant result of this paper.

The present study is not without its own limitations, the most important stemming from its intrinsic nature. When dealing with studies that analyze secondary sources, it is not possible to add new variables or categories, nor to further explore other sources potentially relevant to the study; rather the only material available is what was studied at the time. In the present case, for example, the studies analyzed did not all include questions related to the behavior of bystanders, nor were the options provided the same as those from other studies or surveys on the same topic (i.e., León et al. [12] or Fundación FEDE [44]) and the surveys used in this study only included three active responses; moreover, the option 'other' was used in some cases with no specification of what that includes; and in some cases (i.e., Table 4) the number of subjects in some categories was very small (with a significant number of cells containing less than 20 subjects), which implies the need to interpret the data obtained with caution. Additionally, another limitation that should be noted is the lack of control for some potential confounders

not addressed by the analytical approach developed (e.g., the relationship between age or sex and the analyzed variables can be moderated by some other variables such as level of education).

Further study is required for some issues arising from the data analyzed such as, for example, the important differences related to communicating the violence to close family and friends, or the response of these people and their involvement to a greater or lesser degree with the previous or current IPVAW. Additionally, some sociological surveys analyzed are quite old (more than 10 years) and attitudes, perceptions and behaviours have changed. In fact, another point in need of further analysis is the possible "pandemic effect" brought about by Covid-19 and the restrictive measures taken to control this epidemic [82]. This effect can already be noted in data related to reported cases [83], but further analysis is required on its effect on the bystander response [37]. Additionally, a future meta-analysis, pooling data from multiple studies (publicly available data sets, but also peer-reviewed articles and grey literature), could be done to provide more precise data on the volume of bystanders in cases of IPVAW in Spain.

Despite these limitations, the present study meets its intended objective and has enabled us to know that, in fact, in the cases of IPVAW there are persons within the victim's inner circle who are firsthand witnesses or have been informed by the victim of the existence of this type of violence and, although these bystanders claim they would engage in an active and supportive response, this is in fact not always the case. This underscores the need to develop intervention programs aimed at IPVAW bystanders to improve their reaction and contribute to the development of helpful and efficient active responses, thus activating mechanisms to adequately protect the victims and contribute to the secondary and tertiary prevention of a violence that constitutes a social and health problem of epidemic proportion.

## Acknowledgments

We would like to thank the University of Balearic Islands and the Pontifical University of Salamanca for their support in conducting this research.

## Author Contributions

**Conceptualization:** Andrés Sánchez-Prada, Carmen Delgado-Alvarez, Esperanza Bosch-Fiol, Victoria A. Ferrer-Perez.

**Data curation:** Andrés Sánchez-Prada, Carmen Delgado-Alvarez, Victoria A. Ferrer-Perez.

**Formal analysis:** Victoria A. Ferrer-Perez.

**Funding acquisition:** Victoria A. Ferrer-Perez.

**Investigation:** Virginia Ferreiro-Basurto, Victoria A. Ferrer-Perez.

**Methodology:** Andrés Sánchez-Prada, Carmen Delgado-Alvarez.

**Project administration:** Victoria A. Ferrer-Perez.

**Software:** Virginia Ferreiro-Basurto.

**Supervision:** Victoria A. Ferrer-Perez.

**Validation:** Victoria A. Ferrer-Perez.

**Writing – original draft:** Andrés Sánchez-Prada, Carmen Delgado-Alvarez, Esperanza Bosch-Fiol, Virginia Ferreiro-Basurto, Victoria A. Ferrer-Perez.

**Writing – review & editing:** Andrés Sánchez-Prada, Carmen Delgado-Alvarez, Esperanza Bosch-Fiol, Virginia Ferreiro-Basurto, Victoria A. Ferrer-Perez.

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
