## [Decision Letter · Decision Letter 0]

27 Apr 2022

PONE-D-22-02976Bystanders of intimate partner violence against women and their willingness to intervene: an analysis of secondary data in Spain (2005-2020)PLOS ONE

Dear Dr. Ferrer-Perez,

Thank you for submitting your manuscript to PLOS ONE. After careful consideration, we feel that it has merit but does not fully meet PLOS ONE’s publication criteria as it currently stands. Therefore, we invite you to submit a revised version of the manuscript that addresses the points raised during the review process.

We look forward to receiving your revised manuscript.

Kind regards,

Alfonso Arteaga

Academic Editor

PLOS ONE

Journal Requirements:

4. Please include a copy of Table 11 which you refer to in your text on page 23.

5. We note you have included a table to which you do not refer in the text of your manuscript. Please ensure that you refer to Table 2 in your text; if accepted, production will need this reference to link the reader to the Table.

Additional Editor Comments:

This paper addresses a matter of great interest: the prevention of violence against women in intimate partner relationships knowing the behavior and willingness to intervene of bystanders. The approach of the study, getting valuable information regarding IPVAW from macro-surveys, enriches the knowledge in this field.

However, the manuscript presents important limitations that should be revised and amended. The paper advances knowledge very little, as it synthesizes what is already described in the reports of the surveys, without providing a theoretical analysis and hypotheses based on previous scientific literature. A deeper discussion based on existing knowledge is needed to make a real good contribution to this field of knowledge.

The Introduction section lacks a complete revision of previous research. As pointed by reviewer 2, this section must show what we know about this topic, what is consistent and inconsistent in previous research, and how the current study contributes to addressing these gaps. The article does not have a strong theoretical foundation and lacks specific hypotheses (it is not well justified why it is not necessary to state hypotheses). The frame of the study must be elaborated and this whole section redone.

Due to the multiple secondary studies used, the paper needs a reorganization of the information, which should be organized and synthetized in a clearer way (reviewer 2 recommends a summary table, what may be a good idea). The paper presents too many figures, so authors should revise them and reorganize the information. In addition, as reviewers agree, further information about the variables, the question format, etc., should be provided.

The discussion section must be deeply revised. No strong interpretations of results are given based on existing knowledge in this field. Therefore, this section should consider and be based on the revised Introduction section that must be done.

Several methodological changes must be introduced so that the paper may be suitable for publication. Authors must follow indications of reviewers, as a condition to reconsider the manuscript for revision and possible publication. A major revision must be carried out taking into account all reviewers’ comments.

Reviewers' comments:

Reviewer's Responses to Questions

**Comments to the Author**

1. Is the manuscript technically sound, and do the data support the conclusions?

Reviewer #1: Partly

Reviewer #2: Partly

Reviewer #3: Yes

2. Has the statistical analysis been performed appropriately and rigorously? 

Reviewer #1: No

Reviewer #2: No

Reviewer #3: Yes

3. Have the authors made all data underlying the findings in their manuscript fully available?

Reviewer #1: Yes

Reviewer #2: Yes

Reviewer #3: Yes

4. Is the manuscript presented in an intelligible fashion and written in standard English?

Reviewer #1: Yes

Reviewer #2: Yes

Reviewer #3: Yes

5. Review Comments to the Author

Reviewer #1: The manuscript entitled 'Bystanders of intimate partner violence against women and their willingness to intervene: an analysis of secondary data in Spain (2005-2020)' deals with the issue of prevention of violence against woman in intimate partner relationships taking into account the immediate environment of the victim, through the analysis of data obtained in various surveys conducted in Spain over 15 years. Given that encouraging the helping behavior of the victim's environment is a prevention strategy that has been considered essential, knowing the behavior and willingness to intervene of the bystanders is of great interest. However, there are some important limitations to note in the manuscript.

First, the introduction does not provide an adequate background of previous scientific literature. In fact, what the authors do is a very detailed description of the legislative framework and the surveys carried out in Spain in a theoretical vacuum and a marked absence of previous literature on the subject. In short, they do not present an adequate 'state of the art' to understand the results of the study or how this study advances existing knowledge in this field. There are a significant number of previous studies in this area that are not mentioned or taken into account (e.g., Felson & Paré, 2005; Gracia & Herrero, 2006; Kim & Ferraresso, 2021; Lazarus & Signal, 2013). This section should be redone, framing the study appropriately and presenting the legislative and survey description in Spain in a much shorter way, in a sub-section reflecting the 'present study'.

In the design section, the authors indicate that they will use a descriptive and associative analysis strategy. They also point out that for descriptive analyses it is not necessary to state hypotheses, but what happens with associative analyses? Based on previous scientific literature, the type of relationships expected to be found between variables and in which direction should be proposed.

A more detailed description of how variables are measured should be given in the variables section.

In general, the results described in the paper advance knowledge very little, as they lack a theoretical analysis and hypotheses based on previous scientific literature, beyond synthesizing what is already described in the reports of the surveys themselves and which the reader can find directly in these sources of information. Although some comparative analyses are made according to sex and age, no deeper interpretations of these results are given based on existing knowledge in this field.

Felson, R. B., & Paré, P. P. (2005). The reporting of domestic violence and sexual assault by nonstrangers to the police. Journal of Marriage and Family, 67(3), 597-610.

Gracia, E., & Herrero, J. (2006). Public attitudes toward reporting partner violence against women and reporting behavior. Journal of Marriage and Family, 68(3), 759-768.

Kim, C., & Ferraresso, R. (2021). Factors associated with willingness to report intimate partner violence (IPV) to police in South Korea. Journal of Interpersonal Violence, Doi: 0886260521990837.

Lazarus, K., & Signal, T. (2013). Who will help in situations of intimate partner violence: Exploring personal attitudes and bystander behaviours. International Journal of Criminology and Sociology, 2, 199-209.

Reviewer #2: This article addresses an important topic and has the potential to make a strong contribution to the literature. There are, however, a number of issues that need to be addressed and ways in which the article could be strengthen.

1) Introduction: The authors do a very good job contextualising the study and providing information about IPV in Spain. They, however, do not appropriately discuss previous research (e.g., what do we know about this topic, what is consistent and inconsistent in previous research, and how the current study contributes to addressing these gaps). I think this is particularly important when it comes to justifying the selection of age and gender as key variables for the analysis (instead of education, income, or any other variable). The article at the moment does not have a strong theoretical foundation and lacks specific hypotheses. I encourage the authors to review the front-end of their manuscript to make it more theoretically driven and better justify their methodological decisions. In addition to this, please review this section to fix some typos/grammatical issues and amend the acronym used when referring to the Centro de Investigaciones Sociológicas (as shown on their website, they maintain their Spanish name and acronym in English: https://www.cis.es/cis/opencms/EN/8_cis/quienessomos/).

2) Methods: This section is a bit hard to follow at the moment given the multiple secondary studies used. I would recommend creating a summary table that provides information for each study (organism in charge of data collection, population of interest, data collection dates, language(s) in which the survey was administered, mode(s) of data collection, achieved sample size, response rates, and any other relevant information). I would like to see a similar table with the question wording for each of the variables, and further information about the question format (at the moment it is hard to know whether some questions were closed- or open-ended, if they were exclusive or mark-all-that-apply, but these are the main variables of the study and clarity is paramount here).

I am also concerned that there is no mention, in the "Data Analysis" section to the weights used in the analysis to account for the sampling design and nonresponse. To my knowledge, the CIS provides them and they should be used to generate figures/conduct analysis. Could the authors confirm that they used the weights, and if not, update their analyses to incorporate them when available?

I would also encourage the authors to indicate the software that they used to conduct the analyses, as this is best practice for replicability.

With regards to the files, I do not think that providing combined data for a 10-year period, and then separate data for 2020 is the best approach. I would report the data by year to identify any relevant patterns over time rather than combining the first 10 years. This would not only provide more granular information, but would also help to assess whether changes with respect to 2020 reflect previous patterns or point to something else.

3) Results: Were any corrections used to account for the multiple comparisons? Because of the many bivariate tests that you conduct, the risk of Type I error is high and it would be helpful to describe how this has been minimised. I would also like to see effect sizes in addition to p-values to better assess the magnitude of the differences. With X2, for example, you can use Cramer´s V. I also suggest proofreading this section to amend some typos.

Currently there are too many figures. I would encourage the authors to consolidate their figures and find ways to combine the information. The results section is also a bit repetitive as the findings are disaggregated by sex, by age, and then by sex and age. In terms of the figures, I think they can be made more self-explanatory with a few tweaks. For example, it is unclear to me what the arrows represent and I could not find an explanation on the Figures themselves. The sample size should be specified, and the bars should be accompanied by confidence intervals, to reflect the uncertainty around the estimates.

In the case of Figures 12 and 14, readability is quite poor, given that there is little differentiation across categories. I encourage the authors to explore other alternatives to present their data.

With the youth study (2992), what is the rationale for the age grouping? 15-17 seems much more restricted that 18-29. Were the groups balanced in terms of sample size? Perhaps creating further groups (18-20, 21-23...) would be preferred here.

When presenting the findings from the macro surveys, because you´re using sub-samples (e.g., only women who reported IPV, only women in current relationships) it is essential to provide the sample sizes in all cases. As currently worded, reporting percentages only, it is not always possible for readers to see how many cases were used to compute the figures provided. As shown in Table 1, some of the cells have very few cases (<20). Because of this, I would encourage the authors to be more cautious when interpreting the findings, reminding readers that some of the findings are based on very small sample sizes. This would be something helpful to discuss among the limitations of the study.

As I suggested earlier, I would report the data for each year in Table 2, so that readers have access to all of the information and can better assess trends, rather than getting only three data points and assuming linearity in between.

4) Discussion: I am not convinced that the data used here allows you to estimate "the volume of bystanders in cases of IPVAW in Spain". In the case of the early studies, to the best of my knowledge, only a dichotomous question is used, so a person answering positively could be aware of one or multiple cases of IPVAW. In the case of the macro-surveys, they only account for cases in which the victims actively reported the cases to other people, but fail to account for other sources by which people could be aware of such cases (e.g., neighbours watching or hearing violent events). I think, at best, you have an estimate that underrepresents actual awareness, but I don't think this is properly discussed or addressed in your article.

When discussing awareness differences across age groups, you attribute them to generational effects ("the data obtained underscore the existence of a certain generational effect"). However, I don't think the data used here allow you to disentangle ageing, period, and cohort effects. I encourage you to revise the wording to ensure that the interpretation is fully consistent with the analytical approach and the data used.

In addition to this, I think some limitations of the study are not appropriately discussed. For example, some of the studies are quite old (>10 years) and attitudes, perceptions and behaviours might have changed since. I would also like to see further discussion of the implications of the study (how can these findings inform future research and practice?, how can we better measure this going forward?, what strategies can be used to mitigate social desirability biases in survey research?).

I wish the authors all the best moving this article forward!

Reviewer #3: This paper offers a macro-study with survey data on social perception of gender violence and the 2014 and 2019 macro-surveys). This research data analyzed that in the cases of IPVAW, there are individuals within the victim’s inner circle who are direct witnesses or have been informed by the victim of the existence of this type of violence, and, although these bystanders claim they would engage in a supportive response, this is not always the case.

I would like to congratulate the authors for their work which approaches a very relevant topic nowadays and allows other researchers to have valuable information regarding IPVAW from macro-surveys, which undoubtedly, enriches our knowledge in this field. It would be a reference for subsequent researchers who publish future papers on this topic. In addition, the manuscript is very well-written and methodologically sound.

Just some minor aspects to revise:

Comment 1. In the introduction section, I suggest expanding the literature on the different helping reactions to IPVAW (e.g., informal helping reactions such as "emotional support to the victim" or formal helping responses such as “report to the police”)

Comment 2. On page 9, related to the variables, give an example of each measure, for example, the knowledge of IPVAW and the type of helping behavior.

Comment 3. In the discussion section, I suggest highlighting the recommendations that intervention programs should be incorporated to improve the bystander's reactions and contribute to developing helpful and efficient active responses.

6. PLOS authors have the option to publish the peer review history of their article (what does this mean?). If published, this will include your full peer review and any attached files.

Reviewer #1: No

Reviewer #2: No

Reviewer #3: No

---

## [Author Response · Author response to Decision Letter 0]

3 Jun 2022

Dear editor, 

According to the editor and reviewers’ suggestions, we have revised and changed our paper “Bystanders of intimate partner violence against women and their willingness to intervene: an analysis of secondary data in Spain (2005-2020)" (PONE-D-22-02976) in order to improve it. Specifically we have made the following changes:

Author’s response: We have ensured that our manuscript meets PLOS ONE's style requirements, including those for file naming. Specifically, we have followed the PLOS ONE style templates (https://journals.plos.org/plosone/s/file?id=wjVg/PLOSOne_formatting_sample_main_body.pdf and https://journals.plos.org/plosone/s/file?id=ba62/PLOSOne_formatting_sample_title_authors_affiliations.pdf) and the rules related to tables (https://journals.plos.org/plosone/s/tables) and figures (https://journals.plos.org/plosone/s/figures). 

It should be noted that in this case (changes in document format or in Tables format) we have not used the tool "track changes" option in Microsoft Word for avoiding mistakes and too much coloured text.

In the case of the numbered list of references, the numbering has been modified to adapt it to the current references, presenting only the correct numbering (the original numbering has not been kept for clarity). To make them easier to identify, the new references introduced in response to the reviewers' suggestions have been marked as an insertion.

2. In your Data Availability statement, you have not specified where the minimal data set underlying the results described in your manuscript can be found. 

Author’s response: The methodological strategy used was an analysis of secondary data obtained and processed by other sources. In this case, quantitative data taken from sociological surveys conducted in Spain by different institutions and administrative data related to the records of complaints for intimate partner violence against women and they are fully available on the web freely for anyone who wishes to consult them.

The URLs from the Centro de Investigaciones Sociologicas (https://www.cis.es/cis/opencms/EN/index.html) where the quantitative data from sociological surveys are available are: 

- Social perception of gender violence (study 2968): https://www.cis.es/cis/opencm/ES/1_encuestas/estudios/ver.jsp?estudio=14086

- Social perception of gender violence among adolescents and youth (study 2992): https://www.cis.es/cis/opencm/ES/1_encuestas/estudios/ver.jsp?estudio=14106

- Macro-survey on violence against women 2014 (study 3027): https://www.cis.es/cis/opencm/ES/1_encuestas/estudios/listaMuestras.jsp?estudio=14084

- Macro-survey on violence against women 2019 (study 3235): https://www.cis.es/cis/opencm/ES/1_encuestas/estudios/listaMuestras.jsp?estudio=14470

The URL from the Statistics Portal of the Spanish Government Delegation for Gender Violence where the administrative data related to the records of complaints for intimate partner violence against women are available are: http://estadisticasviolenciagenero.igualdad.mpr.gob.es/

All these data are fully anonymized.

3. We note that you have stated that you will provide repository information for your data at acceptance. 

Author’s response: Apologies for the mistake.

All data files are available from the Centro de Investigaciones Sociologicas (https://www.cis.es/cis/opencms/EN/index.html) and from the Statistics Portal of the Spanish Government Delegation for Gender Violence (http://estadisticasviolenciagenero.igualdad.mpr.gob.es/)

4. Please include a copy of Table 11 which you refer to in your text on page 23.

Author’s response: Apologies for the mistake, which has been corrected in the text.

On line 654 it said "see Table 11", when it should have said "see Table 2" (now, with changes introduced it says “see Table 5”).

5. We note you have included a table to which you do not refer in the text of your manuscript. Please ensure that you refer to Table 2 in your text; if accepted, production will need this reference to link the reader to the Table.

Author’s response: Apologies for the mistake, which has been corrected in the text.

On line 654 it said "see Table 11", when it should have said "see Table 2" (now, with changes introduced it says “see Table 5”).

Financial disclosure.

Author’s response: According to the instructions of Spanish Minister, the disclosure sentence may be:

This work was financed by the Spanish State Research Agency (Agencia Estatal de Investigación), through the Research Project PID2019-104006RB-I00/MCIN/AEI/10.13039/501100011033

Additional Editor Comments

This paper addresses a matter of great interest: the prevention of violence against women in intimate partner relationships knowing the behavior and willingness to intervene of bystanders. The approach of the study, getting valuable information regarding IPVAW from macro-surveys, enriches the knowledge in this field.

However, the manuscript presents important limitations that should be revised and amended. The paper advances knowledge very little, as it synthesizes what is already described in the reports of the surveys, without providing a theoretical analysis and hypotheses based on previous scientific literature. A deeper discussion based on existing knowledge is needed to make a real good contribution to this field of knowledge.

Authors' reply: We have taken all the reviewers' suggestions into account and have carried out a major revision of the paper.

The Introduction section lacks a complete revision of previous research. As pointed by reviewer 2, this section must show what we know about this topic, what is consistent and inconsistent in previous research, and how the current study contributes to addressing these gaps. The article does not have a strong theoretical foundation and lacks specific hypotheses (it is not well justified why it is not necessary to state hypotheses). The frame of the study must be elaborated and this whole section redone.

Authors' reply: We have taken all the reviewers' suggestions into account and have carried out a major revision of the introduction section. The specific response to the reviewer indicates the changes made and where they are placed in the manuscript.

Due to the multiple secondary studies used, the paper needs a reorganization of the information, which should be organized and synthetized in a clearer way (reviewer 2 recommends a summary table, what may be a good idea). 

Authors' reply: We have taken all the reviewers' suggestions into account and have carried out a major revision of the method section (including a summary table of the secondary studies used). The specific response to the reviewer indicates the changes made and where they are placed in the manuscript.

The paper presents too many figures, so authors should revise them and reorganize the information. 

Authors' reply: We have taken all the reviewers' suggestions into account and reduced the figures (from 14 to 9) and reorganized the information.

In addition, as reviewers agree, further information about the variables, the question format, etc., should be provided.

Authors' reply: We have taken all the reviewers' suggestions into account and have carried out a major revision of the method section (including a table with the variables analysed). The specific response to the reviewer indicates the changes made and where they are placed in the manuscript.

The discussion section must be deeply revised. No strong interpretations of results are given based on existing knowledge in this field. Therefore, this section should consider and be based on the revised Introduction section that must be done.

Authors' reply: We have taken all the reviewers' suggestions into account and have carried out a major revision of the discussion section. The specific response to the reviewer indicates the changes made and where they are placed in the manuscript.

Referees requirements:

Referee 1:

The manuscript entitled 'Bystanders of intimate partner violence against women and their willingness to intervene: an analysis of secondary data in Spain (2005-2020)' deals with the issue of prevention of violence against woman in intimate partner relationships taking into account the immediate environment of the victim, through the analysis of data obtained in various surveys conducted in Spain over 15 years. Given that encouraging the helping behavior of the victim's environment is a prevention strategy that has been considered essential, knowing the behavior and willingness to intervene of the bystanders is of great interest. However, there are some important limitations to note in the manuscript.

Author’s response: As reviewer 1 points out, “Given that encouraging the helping behavior of the victim's environment is a prevention strategy that has been considered essential, knowing the behavior and willingness to intervene of the bystanders is of great interest”.

That is precisely why, as we pointed out in the summary of the article (lines 28-30; lines 199-202), the objective of our study is to estimate the volume of bystanders in cases of IPVAW in Spain between 2005 and 2020 (since the entry into force of Organic Law 1/2004 to the present), their willingness to intervene and, in the case of intervention, the type of helping behavior (real or hypothetical) preferred by them. This objective leads us to focus on whether there are witnesses in cases of violence against women (VAW) in Spain and what we know about their behaviour. This research is therefore a preliminary step towards understanding the state of the art (are there VAW witnesses in Spain? How do they behave?), and, on this basis, to be able to design more effective intervention programs to prevent VAW in the future.

A scoping review on bystander helping behaviours in VAW cases in Spain showed us that there was little previous research on this topic in Spain, and, consequently, little information. Precisely, one of the sources of information available and reliable were some sociological surveys that show how, although VAW is often seen as a hidden problem, there are in fact bystanders to this violence. That is why this study uses these sources: survey data (as the surveys of social perception of gender violence and the 2014 and 2019 macro-surveys), and administrative data (as the database of reports filed).

In summary, it is important to note that the objective of our study is not to “deal[s] with the issue of prevention of violence against woman in intimate partner relationships taking into account the immediate environment of the victim”; rather it is to reflect on the bystander behavior and their willingness to intervene in cases of VAW in Spain.

Related to this, we have improved the description of the objective of the study in order to better identify it and avoid any possible confusion in this regard (see lines 199-212).

First, the introduction does not provide an adequate background of previous scientific literature. In fact, what the authors do is a very detailed description of the legislative framework and the surveys carried out in Spain in a theoretical vacuum and a marked absence of previous literature on the subject. In short, they do not present an adequate 'state of the art' to understand the results of the study or how this study advances existing knowledge in this field. There are a significant number of previous studies in this area that are not mentioned or taken into account (e.g., Felson & Paré, 2005; Gracia & Herrero, 2006; Kim & Ferraresso, 2021; Lazarus & Signal, 2013). This section should be redone, framing the study appropriately and presenting the legislative and survey description in Spain in a much shorter way, in a sub-section reflecting the 'present study'.

Author’s response: Thank you to the reviewer 1 for their suggestions. In fact, as reviewer 1 points out, the introduction did not present an adequate 'state of the art' to understand the existing knowledge in the field of bystander behaviour in cases of VAW in Spain.

Therefore, taking into account the recommendation of the reviewer 1, the following has been done:

- The explanation about the VAW legislative situation in Spain has been reduced (we have eliminated the lines 85-111).

- The detailed explanation about sociological surveys in Spain has been removed (we have eliminated the lines 112-172). This explanation has been moved to the method section and it is described following the recommendations of reviewer 2.

- A description on the state of the art regarding the object of study (the witnesses in VAW cases in Spain and their behaviour) has been added (see lines 173-193).

We believe that all these changes now provide an adequate background of previous scientific literature on the topic.

In the design section, the authors indicate that they will use a descriptive and associative analysis strategy. They also point out that for descriptive analyses it is not necessary to state hypotheses, but what happens with associative analyses? Based on previous scientific literature, the type of relationships expected to be found between variables and in which direction should be proposed.

Author’s response: According to the reviewer suggestion, we have added the relationships expected to be found between variables (see lines 231-234).

A more detailed description of how variables are measured should be given in the variables section.

Author’s response: According to the reviewers 1 and 2 suggestion, we have included a new table (Table 2, see line 309) with all this information about each the variables analysed (the variable, the question, and the question format).

In general, the results described in the paper advance knowledge very little, as they lack a theoretical analysis and hypotheses based on previous scientific literature, beyond synthesizing what is already described in the reports of the surveys themselves and which the reader can find directly in these sources of information. Although some comparative analyses are made according to sex and age, no deeper interpretations of these results are given based on existing knowledge in this field.

Author’s response: According to the reviewer’s 1 and 3 suggestions, we have remarked the evolution of the estimates on the volume of bystanders in Spain across last years (including the results of this paper) (lines 675-703). Certainly, some of the surveys analyzed are old, but the results they provide offer more complete information on the subject. Moreover, the 4 sociological surveys provide additional valuable information on the (hypothetical) responses of witnesses in IPVAW cases, which offer us the most important result of this work: there is a significant gap between what witnesses say they will do and what they actually do (as shown by the data on complaints). Certainly, social desirability is one of the problems we encounter, but not the only one. We understand that the main result of this paper is precisely to highlight these contradictions between what is said will be done and what is actually done and stated some of the possible explanations, opening the way for its in-depth study in future works.

Felson, R. B., & Paré, P. P. (2005). The reporting of domestic violence and sexual assault by nonstrangers to the police. Journal of Marriage and Family, 67(3), 597-610.

Gracia, E., & Herrero, J. (2006). Public attitudes toward reporting partner violence against women and reporting behavior. Journal of Marriage and Family, 68(3), 759-768.

Kim, C., & Ferraresso, R. (2021). Factors associated with willingness to report intimate partner violence (IPV) to police in South Korea. Journal of Interpersonal Violence, Doi: 0886260521990837.

Lazarus, K., & Signal, T. (2013). Who will help in situations of intimate partner violence: Exploring personal attitudes and bystander behaviours. International Journal of Criminology and Sociology, 2, 199-209.

Auhor’s response: Thank you for the suggestions. We have added all these references.

Referee 2:

Introduction: 

They, however, do not appropriately discuss previous research (e.g., what do we know about this topic, what is consistent and inconsistent in previous research, and how the current study contributes to addressing these gaps). I think this is particularly important when it comes to justifying the selection of age and gender as key variables for the analysis (instead of education, income, or any other variable). 

Auhor’s response: According to the reviewer suggestion, we have included a discussion about the previous research on the topic and the main results obtained (see lines 173-193).

The article at the moment does not have a strong theoretical foundation and lacks specific hypotheses. I encourage the authors to review the front-end of their manuscript to make it more theoretically driven and better justify their methodological decisions.

Auhor’s response: According to the reviewer suggestion, we have added specific hypothesis and a theoretical foundation related to the relevance of bystander behaviour to improve the IPVAW prevention (see lines 64-65 and lines 231-234).

In addition to this, please review this section to fix some typos/grammatical issues and amend the acronym used when referring to the Centro de Investigaciones Sociológicas (as shown on their website, they maintain their Spanish name and acronym in English: https://www.cis.es/cis/opencms/EN/8_cis/quienessomos/).

Author’s response: According to the reviewer suggestion, we have revised the all section and also we have amended the acronym used to the Centro de Investigaciones Sociológicas (CIS) in all the text.

2) Methods: 

This section is a bit hard to follow at the moment given the multiple secondary studies used. I would recommend creating a summary table that provides information for each study (organism in charge of data collection, population of interest, data collection dates, language(s) in which the survey was administered, mode(s) of data collection, achieved sample size, response rates, and any other relevant information).

Author’s response: According to the reviewer suggestion, we have included a new table (Table 1, see line 243) with all this information about the four sociological surveys analysed.

I would like to see a similar table with the question wording for each of the variables, and further information about the question format (at the moment it is hard to know whether some questions were closed- or open-ended, if they were exclusive or mark-all-that-apply, but these are the main variables of the study and clarity is paramount here).

Author’s response: According to the reviewer suggestion, we have included a new table (Table 2, see line 309) with all this information about each the variables analysed (the variable, the question, and the question format).

I am also concerned that there is no mention, in the "Data Analysis" section to the weights used in the analysis to account for the sampling design and nonresponse. To my knowledge, the CIS provides them and they should be used to generate figures/conduct analysis. Could the authors confirm that they used the weights, and if not, update their analyses to incorporate them when available?

Author’s response: Effectively, the CIS surveys provide some weighting weights for the representativeness of cases in the sample. We have included this information in Table 1, which describes the characteristics of the sociological surveys analysed.

It is important to note that including these weights for analysing the data in our article is not appropriate for the following reasons:

- The fact sheets for studies No. 2968 and No. 2992 provided by the CIS state "no weighting applicable".

- The fact sheets for studies no. 3027 (2014) and no. 3235 (2019) provide two weights for two types of data treatment: 1) weights "to treat the sample as a whole" and 2) weights "to make estimates for the Autonomous Communities". Neither of these two assumptions applies to our article. Our analyses of these data are carried out descriptively on a sub-sample (women who reported having suffered some type of gender-based violence). But no inference is made about the general population (case 1), nor about differences between autonomous communities (case 2), for which it would be necessary to use representativeness weights as pointed out by reviewer 2.

I would also encourage the authors to indicate the software that they used to conduct the analyses, as this is best practice for replicability.

Author response: According to the reviewer suggestion, we have added the software used to conduct the analyses (see line 330-331).

With regards to the files, I do not think that providing combined data for a 10-year period, and then separate data for 2020 is the best approach. I would report the data by year to identify any relevant patterns over time rather than combining the first 10 years. This would not only provide more granular information, but would also help to assess whether changes with respect to 2020 reflect previous patterns or point to something else.

Author response: According to the reviewer suggestion, we have included in a Table (see Table 5) the data about reports filed by year from 2009 to 2020 (see line 658) and we have analysed all data combined for having a whole comprehension and to try to identify some pattern (see lines 635-657).

3) Results: 

Were any corrections used to account for the multiple comparisons? Because of the many bivariate tests that you conduct, the risk of Type I error is high and it would be helpful to describe how this has been minimised. I would also like to see effect sizes in addition to p-values to better assess the magnitude of the differences. With X2, for example, you can use Cramer´s V. I also suggest proofreading this section to amend some typos.

Author’s response: According to the reviewer suggestion, we have incorporated Cramer's V to give an idea of the size of the effect. 

Regarding the accumulation of type I error due to multiple comparisons with the same data, the Bonferroni correction was applied where appropriate. The p-values have been modified in the text with this adjustment. In those cases where the SPSS program provides a p-value = .000, it has been maintained in the text as "p < 0.001" since the exact p-value is not available and, obviously, any correction applied would still be significant at that value.

Currently there are too many figures. I would encourage the authors to consolidate their figures and find ways to combine the information. The results section is also a bit repetitive as the findings are disaggregated by sex, by age, and then by sex and age. In terms of the figures, I think they can be made more self-explanatory with a few tweaks. For example, it is unclear to me what the arrows represent and I could not find an explanation on the Figures themselves. The sample size should be specified, and the bars should be accompanied by confidence intervals, to reflect the uncertainty around the estimates.

Author’s response: According to the reviewer suggestions, we have made the following changes:

- We have reduced the number of figures (from 14 to 9). We have removed Figure 6 because it provided information that could easily be described in the text. Figures 11, 12, 13 and 14 have been removed because they were unclear. The information in figures 11 and 12 has been included in Table 3, the information in figure 13 has been included in Table 4, and the information in figure 14 has been included in Table 5.

- It was decided to keep figures 1 to 5 and 7 to 10 as they provide essential information for the purpose of the study and it was unclear to include the information they contain in the text.

- We have also included the respective 95% confidence intervals in all figures. Given the doubts expressed by the reviewer, and the additional information provided by the confidence intervals, we have decided to remove the arrows because it could cause some confusion (rather than adding clarity).

- We have included the sample size in each case for a better comprehension of the results obtained.

In the case of Figures 12 and 14, readability is quite poor, given that there is little differentiation across categories. I encourage the authors to explore other alternatives to present their data.

Author’s response: According to the reviewer suggestions, the information in figure 12 has been included in Table 3, and the information in figure 14 has been included in Table 5.

With the youth study (2992), what is the rationale for the age grouping? 15-17 seems much more restricted that 18-29. Were the groups balanced in terms of sample size? Perhaps creating further groups (18-20, 21-23...) would be preferred here.

Author’s response: The reason for creating these two age groups is to be able to compare the results of the 2012 survey with the adult population and the 2013 survey with the young population. In this way, in both cases we had a group of 18-29 year olds and we could make comparisons between the results obtained in both studies.

When presenting the findings from the macro surveys, because you´re using sub-samples (e.g., only women who reported IPV, only women in current relationships) it is essential to provide the sample sizes in all cases. As currently worded, reporting percentages only, it is not always possible for readers to see how many cases were used to compute the figures provided. 

Author’s response: According to the reviewer suggestions, we have included the sample size in each case for a better comprehension of the results obtained.

As shown in Table 1, some of the cells have very few cases (<20). Because of this, I would encourage the authors to be more cautious when interpreting the findings, reminding readers that some of the findings are based on very small sample sizes. This would be something helpful to discuss among the limitations of the study.

Author’s response: According to the reviewer suggestions, we have added this limitation of our study in the in the discussion section.

As I suggested earlier, I would report the data for each year in Table 2, so that readers have access to all of the information and can better assess trends, rather than getting only three data points and assuming linearity in between.

Author’s response: According to the reviewer suggestions, the information in figure 14 has been included in Table 5 (before the changes, figure 2). So, the data about reports filed by year from 2009 to 2020 (see line 658) and we have analysed all data combined for having a whole comprehension and to try to identify some pattern (see lines 635-657).

4) Discussion: 

I am not convinced that the data used here allows you to estimate "the volume of bystanders in cases of IPVAW in Spain". In the case of the early studies, to the best of my knowledge, only a dichotomous question is used, so a person answering positively could be aware of one or multiple cases of IPVAW. In the case of the macro-surveys, they only account for cases in which the victims actively reported the cases to other people, but fail to account for other sources by which people could be aware of such cases (e.g., neighbours watching or hearing violent events). I think, at best, you have an estimate that underrepresents actual awareness, but I don't think this is properly discussed or addressed in your article.

Author’s response: According to the reviewer suggestions, we have included a reflection about the limitations of survey data analysed (see lines 696-703).

When discussing awareness differences across age groups, you attribute them to generational effects ("the data obtained underscore the existence of a certain generational effect"). However, I don't think the data used here allow you to disentangle ageing, period, and cohort effects. I encourage you to revise the wording to ensure that the interpretation is fully consistent with the analytical approach and the data used.

Author’s response: According to the reviewer suggestions, we have avoided the use of the term “generational effect” and pointed out that it could be an “age effect” (see line 716).

In addition to this, I think some limitations of the study are not appropriately discussed. For example, some of the studies are quite old (>10 years) and attitudes, perceptions and behaviours might have changed since. I would also like to see further discussion of the implications of the study (how can these findings inform future research and practice?, how can we better measure this going forward?, what strategies can be used to mitigate social desirability biases in survey research?).

Author’s response: According to the reviewer suggestions, we have included a reflection about the limitations of survey data analysed related to the changes over the time (see lines 675-696).

In fact, at the beginning of the discussion we present the estimates on the volume of bystanders in Spain across last years (including the results of this paper). Certainly, some of the surveys analyzed are old, but the results they provide offer more complete information on the subject. Moreover, the 4 sociological surveys provide additional valuable information on the (hypothetical) responses of witnesses in IPVAW cases, which offer us the most important result of this work: there is a significant gap between what witnesses say they will do and what they actually do (as shown by the data on complaints). Certainly, social desirability is one of the problems we encounter, but not the only one. We understand that the main result of this paper is precisely to highlight these contradictions between what is said will be done and what is actually done and stated some of the possible explanations, opening the way for its in-depth study in future works.

Referee 3:

Comment 1. In the introduction section, I suggest expanding the literature on the different helping reactions to IPVAW (e.g., informal helping reactions such as "emotional support to the victim" or formal helping responses such as “report to the police”).

Author’s response: According to the reviewer suggestion, we have added some more explanations and literature on the different helping reactions to IPVAW (see lines 173-193).

Comment 2. On page 9, related to the variables, give an example of each measure, for example, the knowledge of IPVAW and the type of helping behavior.

Author’s response: As the reviewer 2 suggests, we have included a new table (Table 2, see line 309) with all this information about each of the variables analysed (the variable, the question, and the question format).

Comment 3. In the discussion section, I suggest highlighting the recommendations that intervention programs should be incorporated to improve the bystander's reactions and contribute to developing helpful and efficient active responses.

Author’s response: According to the reviewer suggestions, we have included a reflection about challenges for future bystander intervention programs highlighted by our results (see lines 722-727).

---

## [Decision Letter · Decision Letter 1]

17 Aug 2022

PONE-D-22-02976R1Bystanders of intimate partner violence against women and their willingness to intervene: an analysis of secondary data in Spain (2005-2020)PLOS ONE

Dear Dr. Ferrer-Perez,

Thank you for submitting your manuscript to PLOS ONE. After careful consideration, we feel that it has merit but does not fully meet PLOS ONE’s publication criteria as it currently stands. Therefore, we invite you to submit a revised version of the manuscript that addresses the points raised during the review process.

We look forward to receiving your revised manuscript.

Kind regards,

Alfonso Arteaga

Academic Editor

PLOS ONE

Additional Editor Comments:

The paper has improved considerably, as pointed out by the reviewers. Authors have taken into account most of the reviewers’ comments to improve the article. Therefore, the paper is close to being eligible for publication on PlosOne.

However, there are a number of issues that need to be clarified or corrected in the paper. Reviewer 2 points out several errors, methodological and conceptual issues that must be reviewed and modified. If they are taken into account and incorporated in the last version of the manuscript, it will be suitable for publication.

In summary, a major revision must be carried out taking into account the reviewer’s comments. Please, review and respond to each of them.

Reviewers' comments:

Reviewer's Responses to Questions

**Comments to the Author**

1. If the authors have adequately addressed your comments raised in a previous round of review and you feel that this manuscript is now acceptable for publication, you may indicate that here to bypass the “Comments to the Author” section, enter your conflict of interest statement in the “Confidential to Editor” section, and submit your "Accept" recommendation.

Reviewer #1: All comments have been addressed

Reviewer #2: (No Response)

Reviewer #3: All comments have been addressed

2. Is the manuscript technically sound, and do the data support the conclusions?

Reviewer #1: Yes

Reviewer #2: Partly

Reviewer #3: Yes

3. Has the statistical analysis been performed appropriately and rigorously? 

Reviewer #1: Yes

Reviewer #2: No

Reviewer #3: Yes

4. Have the authors made all data underlying the findings in their manuscript fully available?

Reviewer #1: Yes

Reviewer #2: Yes

Reviewer #3: Yes

5. Is the manuscript presented in an intelligible fashion and written in standard English?

Reviewer #1: Yes

Reviewer #2: No

Reviewer #3: Yes

6. Review Comments to the Author

Reviewer #1: The authors have made an important effort to incorporate the recommendations and suggestions I have made. I believe that the manuscript has improved considerably and can be accepted for publication.

Reviewer #2: Overall:

Throughout the manuscript, there continues to be issues with readability and clarity. I encourage the authors to proofread the paper to fix typos and improve readability.

Although the article´s goal is “to estimate the volume of bystanders in cases of IPVAW in Spain between 2005 and 2020”, I continue to believe that this is not achieved. For this, one would ideally conduct a meta-analysis, pooling data from multiple studies (not only publicly available data sets, but also peer-reviewed articles and grey literature). I think being more precise on what the study achieves would provide clarity to the manuscript.

Abstract & Introduction:

In the abstract, and introduction, when justifying the period of time covered on the study, it is indicated that it spans from 2005 to the present. Because the sources only cover 2005-2020 (rather than 2022), I would suggest revising the wording for accuracy.

I think at times the conclusions go beyond the findings. For example, in the abstract it is said that people close to the victims know of their victimisation, and that “although these bystanders claim they would engage in an active and supportive response, this is in fact not always the case.” The wording makes it sound like there is information from the same individuals on both fronts (their awareness of cases of IPV and their willingness to intervene). However, these are different samples and studies, and I think these type of generalisations should be tempered.

Results:

On Table 2 it is indicated that the first three surveys were conducted using face-to-face personal interviews, while the most recent Macro-encuesta used CAPI (which is also a form of personal interviewing). Did the first three studies used PAPI? I would encourage the authors the specify this, and to also include the website where the data can be downloaded, so that readers have all relevant information.

In the “Data analysis” section, it is indicated that histograms were used, but they seem inappropriate given that all variables are nominal.

In some of the figures (Figure 2, Figure 6…), the legend (women/men) seems to be missing.

At times the findings are quite repetitive, as the results are disaggregated by sex, then by age, and then by sex and age (both). Since the results are consistent, I would move some of this information to the appendix (e.g., Figures 4 and 5, Figures 8 and 9) to avoid repetition and streamline this section (this would also help reduce the still high number of figures and tables in the main text).

Also note that, throughout the paper, you refer to “sex” but the categories that you use (women, men) are characteristic of gender identity rather than sex (usually male female). You might want to review this to ensure internal coherence and consistency with the literature.

Table 3, as currently drafted, can be misleading. Although the n reported in the table title is 1,579, the percentages reported in rows 2-7 are based on those who were aware, which is a much smaller number (n=864). Please, review this to avoid confusion.

I would also suggest revising Tables 3 and 4 (which show very similar content) so that they are presented in the same way (the column headings in Table 3 are row headings in Table 4 at the moment, making the comparison between 2014 and 2019 data difficult).

The authors classify responses encouraging the victim to remain with the perpetrator as “passive”. I understand that they are not active in a positive sense (advising to leave), but they are, indeed, active in a negative sense. For me, inaction or passivity would mean not doing anything, whereas encouragement to remain in the relationship would represent a negative active reaction. Perhaps some additional clarity on why the authors interpreted this as inaction would be helpful here.

On page 36, when reporting estimates of IPVAW from the most recent Macro encuesta, the authors disaggregate the results by women who had previously had a partner and those who currently had it. However, for the latter, they report prevalence estimates for partners and ex-partners. Is the inclusion of ex-partner correct in this context? (e.g., Among women who had a partner at the time of the 381 interview (n = 6,506), 14.7% (n = 958) had suffered physical, sexual and/or psychological violence at the hands of their partner or ex-partner in the 12 months prior to the interview, 2.9% physical and/or sexual violence (n = 191), and 14.5% psychological violence (n = 943).).

Similarly, on page 37, the number of women who reported physical, sexual or emotional violence or fear of their current partner is said to be 592, but in the next paragraph, when referring to the same group, the reported n is 520. Please, carefully review the manuscript to resolve inconsistencies.

In Table 5, is the 89.8% for 2013 (victim) correct? Or should this be 8.9%?

Discussion:

When discussing why bystander responses in this study differ from other studies, the authors might want to mention that this is highly dependant on the responses provided in the surveys. While other studies have offered a wider variety of responses, potentially leading to more selections, the surveys used in this study only included three active responses.

Another limitation that should be noted is the lack of control for potential confounders in the analysis (e.g., the relationship between age or sex and the variables of interest can be moderated by variables such as level of education, and this is something that the analytical approach cannot tell).

Finally, I still feel that the discussion can be strengthen by more clearly stating the contribution of the article and its implications for both research and practice (e.g., the authors mention the limitations of the current variables, but it would be helpful to reflect on the ways forward).

Reviewer #3: I think authors have been addressed all the comments and recommendations. Congratulations, the manuscript has improved.

7. PLOS authors have the option to publish the peer review history of their article (what does this mean?). If published, this will include your full peer review and any attached files.

Reviewer #1: No

Reviewer #2: No

Reviewer #3: No

---

## [Author Response · Author response to Decision Letter 1]

31 Aug 2022

Dear editor, 

Thanks to you and to the reviewer 1 and the reviewer 3 for the positive feedback about our revised paper (PONE-D-22-02976R1).

According to you and Reviewers 2 suggestions, we have revised and changed our paper “Bystanders of intimate partner violence against women and their willingness to intervene: an analysis of secondary data in Spain (2005-2020)" (PONE-D-22-02976R1) in order to improve it. Specifically we have made the following changes:

Additional Editor Comments

The paper has improved considerably, as pointed out by the reviewers. Authors have taken into account most of the reviewers’ comments to improve the article. Therefore, the paper is close to being eligible for publication on PlosOne.

Authors' reply: Thank you for the positive feedback.

There are a number of issues that need to be clarified or corrected in the paper. Reviewer 2 points out several errors, methodological and conceptual issues that must be reviewed and modified. If they are taken into account and incorporated in the last version of the manuscript, it will be suitable for publication.

In summary, a major revision must be carried out taking into account the reviewer’s comments.

Authors' reply: We have taken all the Reviewers 2 suggestions into account and have carried out a major revision of the paper.

We respond to each of them hereafter. All these changes are marked using the tool “track changes” option in Microsoft Word

Referees requirements:

Referee 2:

In general:

Throughout the manuscript, there continues to be issues with readability and clarity. I encourage the authors to proofread the paper to fix typos and improve readability.

Authors' reply: We have carefully revised the text of the article to improve its clarity and readability and to eliminate typographical errors. All changes made are marked in the revised document.

Although the article´s goal is “to estimate the volume of bystanders in cases of IPVAW in Spain between 2005 and 2020”, I continue to believe that this is not achieved. For this, one would ideally conduct a meta-analysis, pooling data from multiple studies (not only publicly available data sets, but also peer-reviewed articles and grey literature). I think being more precise on what the study achieves would provide clarity to the manuscript.

Authors' reply: For improving the clarity about the study achieves, the summary (lines 29-30) and introduction (line 106) have been modified to describe accurately the main aim of our paper.

We thank the reviewer for the idea of doing a meta-analysis to include all the available literature and research and thus reach a more accurate and detailed estimate of the bystanders’ volume in Spain. In fact, we have included this suggestion as a line of future work in the conclusions of our paper (lines 642-645).

Abstract & Introduction:

In the abstract, and introduction, when justifying the period of time covered on the study, it is indicated that it spans from 2005 to the present. Because the sources only cover 2005-2020 (rather than 2022), I would suggest revising the wording for accuracy.

Authors' reply: For improving the accuracy, the summary (line 31) and introduction (line 107) have been modified to match the period of the study conducted (which covers 2005 to 2020, not to the present).

I think at times the conclusions go beyond the findings. For example, in the abstract it is said that people close to the victims know of their victimisation, and that “although these bystanders claim they would engage in an active and supportive response, this is in fact not always the case.” The wording makes it sound like there is information from the same individuals on both fronts (their awareness of cases of IPV and their willingness to intervene). However, these are different samples and studies, and I think these type of generalisations should be tempered.

Authors' reply: The description of the results in the summary has been modified and tempered in order to avoid possible generalisations, because effectively they are not the same samples or the same studies (see changes in lines 35-39).

Results:

On Table 2 it is indicated that the first three surveys were conducted using face-to-face personal interviews, while the most recent Macro-encuesta used CAPI (which is also a form of personal interviewing). Did the first three studies used PAPI? I would encourage the authors the specify this, and to also include the website where the data can be downloaded, so that readers have all relevant information.

Authors' reply: On Table 1 (Information about each sociological survey included) we have included the information about mode of data collection exactly as appears in technical sheet of each analysed survey: in the first three surveys the technical sheet says that they were conducted using face-to-face interviews during a home visit, and in the last one the technical sheet says that they was conducted by Computer-assisted personal interviews (CAPI). 

Additionally we have included on the Table 1 the website where the data can be downloaded, as the reviewer suggests (see line 149).

In the “Data analysis” section, it is indicated that histograms were used, but they seem inappropriate given that all variables are nominal.

Authors' reply: The error has been corrected by replacing histogram with bar chart (line 180).

In some of the figures (Figure 2, Figure 6…), the legend (women/men) seems to be missing.

Authors' reply: We have added the legend to the Figures 2 and 6.

At times the findings are quite repetitive, as the results are disaggregated by sex, then by age, and then by sex and age (both). Since the results are consistent, I would move some of this information to the appendix (e.g., Figures 4 and 5, Figures 8 and 9) to avoid repetition and streamline this section (this would also help reduce the still high number of figures and tables in the main text).

Authors' reply: The Figures mentioned by the reviewer refer to the responses that adult men (FIG4), adult women (FIG5), young men (FIG8) and young women (FIG9) probably would give to IPVAW. This is a very relevant information for the objectives of our study and whose analysis is presented in the text and completed with the figures mentioned. 

We understand that if this relevant information is kept as text, but the Figures are moved to the appendix (as suggested by the reviewer) the text does not become clearer if not more confusing. 

However, if the editor considers it convenient, we have no objection to move Figures 4, 5, 8 and 9 to an appendix. We await your instructions in this regard.

Also note that, throughout the paper, you refer to “sex” but the categories that you use (women, men) are characteristic of gender identity rather than sex (usually male female). You might want to review this to ensure internal coherence and consistency with the literature.

Authors' reply: We have revised the Figures and we have changed the legend women – men by female – male to be consistent with the use of the category sex.

Table 3, as currently drafted, can be misleading. Although the n reported in the table title is 1,579, the percentages reported in rows 2-7 are based on those who were aware, which is a much smaller number (n=864). Please, review this to avoid confusion.

Authors' reply: Effectively, we have corrected the n because data in Table 3 are not related to 1,579 women.

Regarding the macro-survey conducted in 2014, among the women interviewed who had been in an intimate partner relationship at some point in their life (n = 9,807), 16.1% (n = 1,579) had been victim of physical violence and/or sexual violence or had been afraid of their partner or ex-partner. And 81% of the women who had suffered some form of IPVAW (n = 1,279) had explained their situation to another person. Table 3 refers to these 1,279 women, and we have changed the n value in this sense (sse line 355 and line 360).

I would also suggest revising Tables 3 and 4 (which show very similar content) so that they are presented in the same way (the column headings in Table 3 are row headings in Table 4 at the moment, making the comparison between 2014 and 2019 data difficult).

Authors' reply: To facilitate comparison between Tables 3 and 4, the Table 3 format has been modified (swapping rows and columns), as suggested by the reviewer.

The authors classify responses encouraging the victim to remain with the perpetrator as “passive”. I understand that they are not active in a positive sense (advising to leave), but they are, indeed, active in a negative sense. For me, inaction or passivity would mean not doing anything, whereas encouragement to remain in the relationship would represent a negative active reaction. Perhaps some additional clarity on why the authors interpreted this as inaction would be helpful here.

Authors' reply: In the introduction (lines 56 to 61) a description was given of what the literature on the subject considers as active/passive responses of bystanders:

"As in the case of other emergency situations, bystanders of VAW and IPVAW have the option of being either “passive” or “inactive”, meaning they may choose not to become involved, to ignore the situation and/or keep quiet and do nothing, supporting the perpetrator and/or blaming the victim, or “active”, meaning to engage and intervene to help the victim providing different forms of informal support to the them (for example, offering assistance, helping them to make decisions, talking to the them, helping or accompanying her to access support services, or helping her to report the case to the police) and/or stop the violence (for example, taking personal action, confronting the aggressor, requesting legal intervention, or reporting the case to the police ) [10-18], thus contributing, or not, to the prevention of such acts of abuse".

In accordance with this criterion, in the analysis of macrosurveys' results (lines 367 to 371), responses that involve doing nothing, as well as those that involve supporting the perpetrator (i.e., showing indifference or encouraging the woman to give the abuser another chance), have been considered as inactive/passive.

On page 36, when reporting estimates of IPVAW from the most recent Macro encuesta, the authors disaggregate the results by women who had previously had a partner and those who currently had it. However, for the latter, they report prevalence estimates for partners and ex-partners. Is the inclusion of ex-partner correct in this context? (e.g., Among women who had a partner at the time of the 381 interview (n = 6,506), 14.7% (n = 958) had suffered physical, sexual and/or psychological violence at the hands of their partner or ex-partner in the 12 months prior to the interview, 2.9% physical and/or sexual violence (n = 191), and 14.5% psychological violence (n = 943).).

Authors' reply: In order to describe these results, we have followed exactly the criteria established by the Centro de Investigaciones Sociologicas to describe the macrosurveys results obtained. 

In fact, in lines 381-388 we have presented two types of results:

First, they are results related to the women interviewed who had previously had a partner: 

“Among the women interviewed who had previously had a partner (n = 9,218), 33.6% (n = 3,098) suffered physical, sexual and/or psychological (emotional, controlling, financial or fear-inducing violence) abuse at the hands of their partner or ex-partner at some point in their life; 14.8% experienced physical and/or sexual (n = 1,362) violence; and 33.2% suffered psychological trauma (n = 3,056)”. 

Secondly, they are results related to the women interviewed who had a partner at the time of the interview: 

“Among women who had a partner at the time of the interview (n = 6,506), 14.7% (n = 958) had suffered physical, sexual and/or psychological violence at the hands of their partner or ex-partner in the 12 months prior to the interview, 2.9% physical and/or sexual violence (n = 191), and 14.5% psychological violence (n = 943)”.

In each case, we have presented results related to the former and the present partners (as in the originals reports of Centro de Investigaciones Sociologicas and Gender Violence Government Delegation).

Similarly, on page 37, the number of women who reported physical, sexual or emotional violence or fear of their current partner is said to be 592, but in the next paragraph, when referring to the same group, the reported n is 520. Please, carefully review the manuscript to resolve inconsistencies.

Author’s reply: Indeed, the wording of the sentence was misleading and confusing. The sentence has been revised to read as follows:

Lines 593 – 596: Among all women who had suffered physical, sexual or emotional violence or had feared their partner (n = 2,395), 21.7% (n = 520) had reported these aggressions either to the police or directly to the courts (25% in the case of former partners; 5.4% in the case of current partners)

In Table 5, is the 89.8% for 2013 (victim) correct? Or should this be 8.9%? 

Authors' reply: We have corrected the mistake. The data should be 8.9%.

Discussion:

When discussing why bystander responses in this study differ from other studies, the authors might want to mention that this is highly dependant on the responses provided in the surveys. While other studies have offered a wider variety of responses, potentially leading to more selections, the surveys used in this study only included three active responses.

Authors' reply: Thank you for the suggestion that has been included in the discussion (lines 559-561 and 625-626).

Another limitation that should be noted is the lack of control for potential confounders in the analysis (e.g., the relationship between age or sex and the variables of interest can be moderated by variables such as level of education, and this is something that the analytical approach cannot tell).

Authors' reply: Thank you for the suggestion that has been included in the discussion (lines 630-633).

Finally, I still feel that the discussion can be strengthen by more clearly stating the contribution of the article and its implications for both research and practice (e.g., the authors mention the limitations of the current variables, but it would be helpful to reflect on the ways forward).

Authors' reply: Thank you for the suggestion that has been included in the discussion (lines 642-645), adding like this some more information to that already included in this regard (lines 646-654).

---

## [Editor Report · Decision Letter 2]

5 Sep 2022

Bystanders of intimate partner violence against women and their willingness to intervene: an analysis of secondary data in Spain (2005-2020)

PONE-D-22-02976R2

Dear Dr. Ferrer-Perez,

We’re pleased to inform you that your manuscript has been judged scientifically suitable for publication and will be formally accepted for publication once it meets all outstanding technical requirements.

Kind regards,

Alfonso Arteaga

Academic Editor

PLOS ONE
---

## [Editor Report · Acceptance letter]

8 Sep 2022

PONE-D-22-02976R2 

Bystanders of intimate partner violence against women and their willingness to intervene: an analysis of secondary data in Spain (2005-2020) 

Dear Dr. Ferrer-Perez:

I'm pleased to inform you that your manuscript has been deemed suitable for publication in PLOS ONE. Congratulations! Your manuscript is now with our production department. 

Kind regards, 

on behalf of

Dr. Alfonso Arteaga 

Academic Editor

PLOS ONE